# AdaptMI: Adaptive Skill-based In-context Math Instructions for Small Language Models

**Yinghui He    Abhishek Panigrahi    Yong Lin    Sanjeev Arora**
Princeton Language and Intelligence, Princeton University
yh0068@princeton.edu

## Abstract

In-context learning (ICL) allows a language model to improve its problem-solving capability when provided with suitable information in context. Since the choice of in-context information can be determined based on the problem itself, in-context learning is analogous to human learning from teachers in a classroom. Recent works (Didolkar et al., 2024a;b) show that ICL performance can be improved by leveraging a frontier large language model's (LLM) ability to predict required *skills* to solve a problem, popularly referred to as an *LLM's metacognition*, and using the recommended skills to construct necessary in-context examples. While this skill-based strategy boosts ICL performance in larger models, its gains on small language models (SLMs) have been minimal, highlighting a performance gap in ICL capabilities.

We investigate this gap and show that skill-based prompting can hurt SLM performance on *easy* questions by introducing unnecessary information, akin to cognitive overload. To address this, we introduce AdaptMI, an **Adapt**ive approach to selecting skill-based in-context **M**ath **I**nstructions for SLMs. Inspired by cognitive load theory from human pedagogy, our method only introduces skill-based examples when the model performs poorly. We further propose AdaptMI+, which adds examples targeted to the specific skills missing from the model's responses. On 5-shot evaluations across popular math benchmarks and five SLMs (1B–7B; Qwen, Llama), AdaptMI+ improves accuracy by up to 6% over naive skill-based strategies. [1]

## 1 Introduction

Human learning is primarily feedback driven (Hattie & Timperley, 2007; Bandura & Walters, 1977). The most common example is how students refine their understanding on a subject through adaptive examples and feedback from a teacher in a classroom setting. In the domain of language models(Vaswani et al., 2017; Achiam et al., 2023; Team et al., 2023; Grattafiori et al., 2024), in-context learning (ICL) (Brown et al., 2020) plays an analogous role. ICL enables models to adapt their problem-solving strategies by conditioning on additional task-relevant information provided in context, possibly sourced from a more capable frontier model acting as a teacher.

However, ICL is known to be an emergent property (Wei et al., 2022), with larger models showing better ICL capabilities than smaller ones. Small Language Models (SLMs) often struggle to generalize from in-context examples and are highly sensitive to how the context is constructed, which limits their ability to learn effectively from in-context instructions. This paper aims to improve the ICL performance of SLMs through careful selection of in-context math instructions.

We build on skill-based in-context example selection from Didolkar et al. (2024a;b). This work leverages the metacognitive abilities of frontier large language models (LLMs) to

---

[1]Code available at: https://github.com/princeton-pli/AdaptMI.

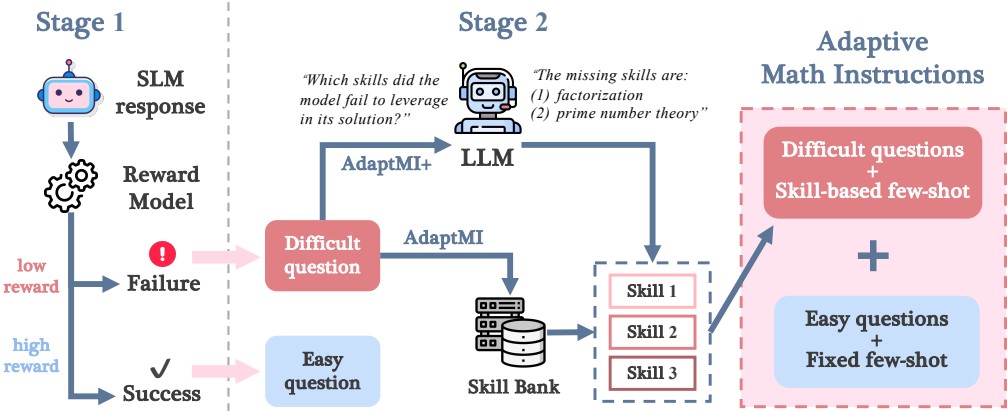

Figure 1: AdaptMI and AdaptMI+ are 2-stage adaptive in-context example selection methods. In the first stage, questions are classified as *easy* and *difficult* using a reward model on the SLM's responses and a threshold-based filtering. In the second stage, AdaptMI uses skill-based in-context examples only for *difficult* questions. For AdaptMI+, we use an LLM to identify the key skills missing in the SLM's responses for *difficult* questions and use specific in-context examples targeted towards the missing skills. For *easy* questions, we use a fixed set of in-context examples. We follow Didolkar et al. (2024a) to get the Skill Bank, skill annotations for each question, and relevant in-context examples for each skill.

predict the high-level skills required to solve a given task. For instance, when presented with a question "3+5*2=", a frontier LLM might infer that the relevant skills are "addition" and "multiplication." After annotating a pool of examples with these skill labels, in-context examples are selected at inference time by first predicting the required skills and then retrieving matching examples. This approach aligns with cognitive theories of human learning that encourage teaching through appropriate skill-based guidance (Kirschner et al., 2006; Sweller, 2011). While skill-based in-context selection significantly boosts the ICL performance of larger models, it fails to improve ICL performance in SLMs.

**Ablation reveals an important insight:** Skill-based strategy can hurt performance of an SLM on *easy* questions—those that an SLM can already solve without skill-based guidance. We define *easy* questions for an SLM using a reward model, illustrated in Figure 1 and formally defined in Section 2.2. Across 5 SLMs on the MATH dataset (Hendrycks et al., 2021), we observe an average 4% performance drop on *easy* questions when using skill-based selection, compared against non skill-based in-context selection strategies. Further fine-grained analysis in Section 4.1 reveals that this strategy leads to long and erroneous responses on *easy* questions, mirroring *overthinking* patterns observed in weaker human students when overloaded with information (Diaconis & Mazur, 2003; Liu et al., 2024b).

**Core Contribution:** Motivated by Adaptive Teaching (Randi, 2022) and Cognitive Load Theory (Sweller, 2011) which suggest humans learn more effectively when guided specifically on tasks they find challenging, we propose AdaptMI. It is an adaptive 2-stage in-context selection method that applies skill-based example selection only to *difficult* questions where the small language model (SLM) struggles. Additionally, mirroring how humans benefit most from feedback on their mistakes, we further propose AdaptMI+, where the specific skills that are missing from the SLM's responses are used to create in-context examples. We provide a method overview in Figure 1, and outline the details in Sections 2.2 and 2.3.

Our experiments (Section 3) on popular math datasets show that AdaptMI+ can significantly improve the performance of all five tested SLMs by up to 6%, while AdaptMI also yields notable gains of up to 3.6%. On top of that, in Section 3.3, we extend AdaptMI+ to an iterative loop of adaptive example selection and demonstrate its potential of progressively, constantly guiding small language models to tackle harder problems. In Section 4, we provide a thorough discussion on why our adaptive example selection method is superior

to naive, non-adaptive skill-based selection. We further present several ablation studies on choices of in-context examples and reward model settings. Finally, we discuss future directions in Section 6.

## 2 Designing AdaptMI and AdaptMI+

### 2.1 Preliminary

We study $k$-shot in-context learning with small language models. Let $\mathcal{Q}$ be the set of evaluation questions, and let $q \sim \mathcal{Q}$ denote a question drawn from this set. We consider the $k$-shot setting, where, given a pool $\mathcal{P}$ of question-answer pairs, $k$ examples are selected and included in the prompt for each evaluation question $q$. There are two common strategies to select the in-context example pairs.

- **Fixed** $k$-shot examples: We fix a set of $k$ examples from $\mathcal{P}$ and use them for inference on all evaluation questions. Our experiments will use the examples used by Qwen models for evaluation (Yang et al., 2024).
- **Random** $k$-shot examples: We utilize $k$ randomly selected examples from the pool of in-context examples $\mathcal{P}$ for each evaluation question.

Our work builds on skill-based in-context selection from Didolkar et al. (2024a), which we describe here. While hard to define precisely, a skill is informally defined as a basic computation necessary to solve a task at hand. For example, necessary skills to solve arithmetic tasks could be addition, subtraction, multiplication and division. We will use Skill-bank($\mathcal{Q}$), as a set of skills that are necessary to solve questions in $\mathcal{Q}$. These skills are enlisted from a large model like GPT-4 using an appropriate prompting strategy(Didolkar et al., 2024a; Kaur et al., 2024). Next, each question in the evaluation set $\mathcal{Q}$ and the in-context example pool $\mathcal{P}$ are matched to the necessary skills from the Skill-bank($\mathcal{Q}$). We will use Skill-Map : $\mathcal{Q} \cup \mathcal{P} \rightarrow$ Skill-bank($\mathcal{Q}$)$^k$ to refer to the map between each question $q$ to a set of $k$ skills, which we will get by prompting an LLM (Achiam et al., 2023). Then, skill-based in-context examples are decided as follows:

- **Skill-based** $k$-shot examples: For each question $q \in \mathcal{Q}$, we pick a set of $k$ examples using Skill-Map($q$), by randomly picking one example for each skill in Skill-Map($q$). More formally, for each skill $s$ in Skill-Map($q$), we randomly pick an example from the pool of in-context examples $\mathcal{P}$ which is annotated with the skill $s$ and return the union of the selected examples for all the skills. This is formally outlined in Algorithm 1 in appendix.

Now, we define AdaptMI and AdaptMI+ built on the above-defined prompting strategies, that consist of 2 primary stages. Section 2.2 formally introduces the first stage that identifies *easy* and *difficult* questions for an SLM using a reward model. Section 2.3 then presents the prompting strategy for AdaptMI and AdaptMI+ on *easy* and *difficult* questions.

### 2.2 Stage 1: Detection of *easy* and *difficult* questions via reward filtering

In this stage, we will label a question $q \in \mathcal{Q}$ as *easy* or *difficult* for an SLM. We could simply define *difficult* questions as those set of questions that the model gets wrong with fixed or random $k$-shot prompting. However, this requires access to the ground truth labels. Instead, to make our technique more broadly applicable, we use a reward model to classify the responses of the SLM. The reward model need not be a perfect reward model, we give more details in Appendix B.1. Given a question $q$, we use a reward model on the response of the SLM when prompted with fixed $k$-shot examples.

**Scoring with a process reward model:** Because we primarily focus on math datasets, we assume that the model's response is composed of $t$ steps for a question $q$ and contains answer in its final step. We will use the reward model to output reward scores for each step. For simplicity, we will refer to the scores of the reward model as $\{r_{q,1}, \cdots, r_{q,t}\}$. Then, we use thresholds $\tau_1, \tau_2$ to filter out *difficult* questions for the SLM. We will refer to the threshold filtering function as $R : \mathcal{Q} \rightarrow \{0, 1\}$.

$$R(q) = \begin{cases} 0, & \text{(if) } r_{q,t} \leq \tau_1 & \text{(final step has low reward)} \\ & \text{(or) } \frac{1}{t}\sum_{i=1}^{t} r_{q,i} \leq \tau_1 & \text{(average low reward across all steps)} \\ & \text{(or) } \exists i < t \text{ s.t. } r_{q,i} \leq \tau_2 & \text{(low reward at any step)} \\ 1, & \text{otherwise,} \end{cases} \quad (1)$$

**Difficult vs. easy questions.** We define $\mathcal{Q}_{\text{difficult}}$ as the set of questions with low-reward model responses $R$. Accordingly, $\mathcal{Q}_{\text{easy}}$ denotes all remaining questions.

$$\mathcal{Q}_{\text{difficult}} = \{q \mid R(q) = 0\}$$
$$\mathcal{Q}_{\text{easy}} = \{q \mid R(q) = 1\} \quad (2)$$

### 2.3 Stage 2: Skill-based selection of in-context examples

AdaptMI uses skill-based $k$-shot examples for *difficult* questions and fixed $k$-shot examples for *easy* questions.

- **AdaptMI**: For *difficult* questions $Q_{\text{difficult}}$, we use skill-based $k$-shot examples. For *easy* question $Q_{\text{easy}}$, we use fixed $k$-shot examples.

Instead of using in-context examples for all skills relevant to a *difficult* question, AdaptMI+ focuses only on the skills that the model's initial response lacks:

- **AdaptMI+**: For each *difficult* question $q$ in $Q_{\text{difficult}}$, we use a large LLM (GPT-4o-mini) to predict the set of skills in Skill-Map$(q)$ that are missing in the model's response. Then, for each skill $s$ that are missing, we randomly pick an example from the pool of in-context examples $\mathcal{P}$ which is annotated with the skill $s$ and return the union of the selected examples for all the missing skills. For *easy* questions $Q_{\text{easy}}$, we use fixed $k$-shot examples.

## 3 Experiment

### 3.1 Experimental Settings

**Datasets.** We evaluate on the MATH (7.5k training samples and 5k test samples) (Hendrycks et al., 2021) and GSM8K (7.4k training samples and 1.3k test samples) (Cobbe et al., 2021) datasets. We follow Didolkar et al. (2024a) to label skills on both the training and test sets using GPT-4o-mini (OpenAI, 2024), and run inference experiments on the whole test set. Appendix A.1 shows the prompt and examples of our skill annotation pipeline. We sample in-context examples from the training set. These two datasets are not overly challenging for SLMs, which ensures relatively interpretable model outputs for stable failure detection. Meanwhile, they are sufficiently representative to offer meaningful insights into our method's efficacy.

**Model settings.** We tested our methods on five instruction-tuned small language models: Qwen2.5-1.5B-Instruct, Qwen2.5-3B-Instruct, Qwen2.5-7B-Instruct, Llama-3.2-1B-Instruct, and Llama-3.2-3B-Instruct (Yang et al., 2024; Meta AI, 2024). We evaluate the models on 5-shot ICL performance. We use generation temperature at 0.0 for all experiments. We also compare against consistency@5 voting (Wang et al., 2022) with 5-shot fixed examples, where we use 5 generations at temperature 1.0 and evaluate the consistent response. For classifying *easy* and *difficult* questions in the first stage, we use RLHFlow/Llama3.1-8B-PRM-Mistral-Data (Xiong et al. (2024)), an 8B process reward model fine-tuned from Llama-3.1-8B, with filtering thresholds $\tau_1 = 0.85, \tau_2 = 0.7$. We use GPT-4o-mini for skill annotation as well as labeling missing skills in AdaptMI+.

**Baselines.** We compare our method to non-adaptive in-context example selection methods, respectively feeding in fixed examples, random examples, and skill-based examples (Didolkar et al. (2024a)) for all queries.

| Methods | MATH | | | | | GSM8K |
| --- | --- | --- | --- | --- | --- | --- |
| | Geometry | Precalculus | Algebra | Prealgebra | **Avg.** | **Avg.** |
| *# Qwen2.5-1.5B-Instruct* | | | | | | |
| Fixed Examples | 39.7 | 38.3 | 72.2 | 67.3 | 52.8 | 71.5 |
| Random Examples | 42.8 | 41.0 | 73.1 | 68.1 | 53.3 | 70.9 |
| Skill-based Examples | 43.2 | 39.6 | 72.0 | 67.7 | 53.0 | 66.1 |
| Consistency@5 | 44.5 | **43.5** | 77.6 | 70.8 | 56.9 | 75.6 |
| **AdaptMI** | **44.7** | 42.1 | 76.8 | 72.0 | 56.4 | 72.9 |
| **AdaptMI+** | 44.5 | 42.1 | **78.2** | **72.8** | **57.2** | **75.8** |
| *# Qwen2.5-3B-Instruct* | | | | | | |
| Fixed Examples | 56.4 | 53.5 | 85.4 | 79.7 | 66.6 | 84.7 |
| Random Examples | 54.7 | 53.7 | 85.3 | 78.9 | 66.1 | 84.9 |
| Skill-based Examples | 53.4 | 55.7 | 86.2 | 80.7 | 66.9 | 85.4 |
| Consistency@5 | **61.9** | 55.3 | 87.4 | 81.4 | 68.9 | 87.0 |
| **AdaptMI** | 54.9 | **56.2** | 87.7 | 81.8 | 67.8 | 87.4 |
| **AdaptMI+** | 56.0 | 55.5 | **88.3** | **82.1** | **69.1** | **87.7** |
| *# Qwen2.5-7B-Instruct* | | | | | | |
| Fixed Examples | 61.2 | 61.5 | 91.2 | 87.1 | 74.7 | 91.7 |
| Random Examples | 60.1 | 62.1 | 91.4 | 86.6 | 74.4 | 91.1 |
| Skill-based Examples | 61.2 | 64.3 | 90.6 | 87.7 | 74.4 | 91.7 |
| Consistency@5 | 62.4 | 57.7 | 92.3 | 87.0 | 75.1 | **93.3** |
| **AdaptMI** | 62.2 | **64.7** | 91.5 | 87.6 | 75.9 | 92.3 |
| **AdaptMI+** | **64.9** | 63.4 | **92.8** | **88.8** | **76.7** | 92.4 |
| *# Llama-3.2-1B-Instruct* | | | | | | |
| Fixed Examples | 8.0 | 11.1 | 19.6 | 21.3 | 13.8 | 26.8 |
| Random Examples | 10.2 | 6.5 | 24.0 | 20.9 | 13.7 | 19.3 |
| Skill-based Examples | 14.8 | 6.8 | 16.7 | 22.6 | 13.4 | 13.4 |
| Consistency@5 | 13.6 | **13.3** | 28.8 | 28.2 | 19.4 | **29.9** |
| **AdaptMI** | 13.6 | 10.3 | 20.8 | 29.3 | 16.2 | 23.2 |
| **AdaptMI+** | **17.1** | 11.1 | **29.6** | **35.4** | **19.8** | 26.0 |
| *# Llama-3.2-3B-Instruct* | | | | | | |
| Fixed Examples | 26.1 | 29.8 | 63.8 | 67.6 | 46.2 | 75.8 |
| Random Examples | 34.1 | 26.9 | 61.9 | 55.3 | 41.3 | 76.2 |
| Skill-based Examples | 29.6 | 31.7 | 66.2 | 63.3 | 45.9 | 71.7 |
| Consistency@5 | **36.1** | 23.9 | 60.0 | 61.9 | 44.1 | **80.7** |
| **AdaptMI** | 28.4 | 31.7 | **71.6** | **71.3** | **49.8** | 76.4 |
| **AdaptMI+** | 29.6 | **35.6** | 68.1 | **71.3** | 49.4 | **80.7** |

Table 1: AdaptMI and AdaptMI+ demonstrate a consistent accuracy gain by up to 3.6% and 6% respectively, compared with baseline methods. We present all results as Pass@1 accuracy unless otherwise indicated. Due to space limits, we provide the results on Number Theory, Intermediate Algebra, and Counting & Probability in Appendix D.

## 3.2 Performances of AdaptMI and AdaptMI+

Table 1 reports the main results of our adaptive in-context learning method. The baseline methods with non-adaptive in-context examples (fixed, random, or skill-based) results in largely similar Pass@1 accuracy, while consistency@5 can improve accuracy by a few percentages. Across all model sizes, our methods AdaptMI and AdaptMI+ consistently outperform the non-adaptive Pass@1 baselines, and are on par with Consistency@5 performance on most subareas. The overall improvements are especially pronounced for smaller models, Qwen2.5-1.5B-Instruct and Llama-3.2-1B-Instruct.

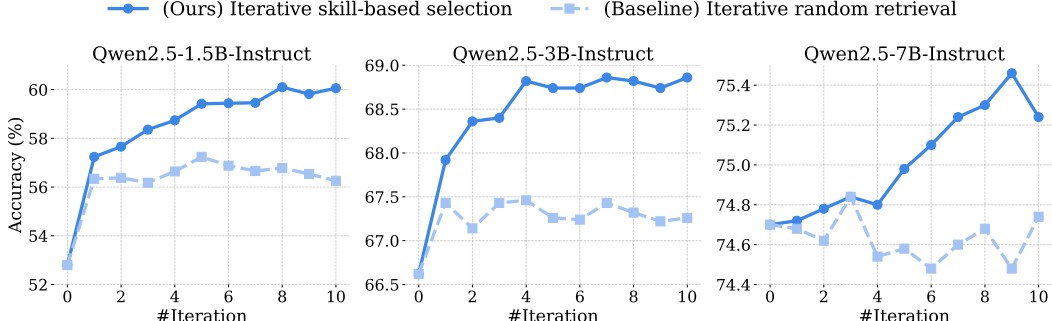

Figure 2: SLM performances under iterative skill-based example selection (AdaptMI+) vs. iterative random example retrieval. Each iteration involves model inference, *difficult* question detection, and random/skill-based example re-selection with GPT-4o-mini. Iterative AdaptMI+ yields a continuous accuracy gain by up to 7.2%, while the baseline leads to fluctuated performances.

While AdaptMI surpasses consistency@5 performance on most domains, it slightly lags behind on certain subjects such as Geometry and Precalculus for 1B or 3B models. These subjects are relatively difficult for the model, as suggested by their loss scores compared to other subjects (see Table 12 in Appendix). Since AdaptMI requires models to have sufficient capabilities to leverage the given skill-based examples, it may not work better than Consistency@5 on these harder topics.

Notably, AdaptMI+ brings significant performance gain across all areas by up to 6%, reflecting its strength in accurately targeting model failures. AdaptMI also substantially improves performance by up to 3.6% for Qwen2.5-1.5B-Instruct, Llama-3.2-1B-Instruct, and Llama-3.2-3B-Instruct on MATH. This indicates that our adaptive instruction methods are effective on lower-performing models even without the aid of an LLM.

On stronger models such as Qwen2.5-3B-Instruct and Qwen2.5-7B-Instruct, however, AdaptMI shows smaller effectiveness compared to AdaptMI+. This may suggest that higher-performing models require a more intelligent and target skill identification process. Overall, these results demonstrate the effectiveness of adaptive example selection and highlight the potential of our approach to elicit the full reasoning capabilities of small language models.

### 3.3   Iterative AdaptMI+

Our method can be extended to an iterative loop of adaptive example selection. Each iteration begins with model inference, followed by detecting *difficult* questions and using GPT-4o-mini to select skill-based examples. The selected examples are then fed in with *difficult* questions for model inference in the next iteration. This iterative AdaptMI+ is essentially pushing the SLM to tackle a gradually refined set of *difficult* questions by adaptive teaching. We compare iterative AdaptMI+ with a baseline of iterative random retrieval, where the loop involves inference, random example resampling, and re-inference.

Figure 2 shows that iterative AdaptMI+ consistently improves the reasoning performance on MATH for all three Qwen small language models, while the baseline method struggles to keep pushing the accuracy boundary after the first few iterations. For 1.5B and 3B models, the performance grows rapidly in the first four iterations, and improves more gradually thereafter. The 7B model performance, while starting to degrade by the 10th loop, still increases substantially compared to baseline. Through iterative re-selection of targeted in-context examples, iterative AdaptMI+ demonstrates the potential of progressively guiding small language models to tackle unsolved problems.

| Question Difficulty | Example Type | MATH | | | | | GSM8K |
|---|---|---|---|---|---|---|---|
| | | Geometry | Precalculus | Algebra | Prealgebra | **Avg.** | **Avg.** |
| Difficult | Fixed | 21.3 | 23.7 | 44.8 | 35.1 | 29.8 | 45.2 |
| | Random | 23.2 | 25.3 | 53.9 | 40.5 | 31.2 | 46.1 |
| | Skill-based | **28.4** | **28.9** | **55.1** | **45.5** | **35.7** | **48.0** |
| | | +7.1 | +5.2 | +10.3 | +10.4 | +5.9 | +2.8 |
| Easy | Fixed | **82.1** | **81.8** | **94.6** | **93.7** | **90.2** | **96.3** |
| | Random | 81.6 | 78.9 | 92.1 | 92.3 | 87.6 | 90.6 |
| | Skill-based | 77.2 | 71.5 | 85.9 | 86.0 | 81.0 | 83.2 |
| | | -4.9 | -10.3 | -8.7 | -7.7 | -9.2 | -13.1 |

Table 2: Accuracy of Qwen2.5-1.5B-Instruct on *difficult* and *easy* questions, respectively under fixed, random, and skill-based examples. Skill-based examples boost performance on *difficult* questions across all categories, while significantly underperforming on *easy* questions. We provide the results on Number Theory, Intermediate Algebra, and Counting & Probability, as well as the results on other Qwen models in Appendix D.

## 4 Discussion

### 4.1 Why does adaptive selection work better than non-adaptive skill-based selection?

To better understand, we compare performance under fixed, random, and skill-based in-context examples on *easy* and *difficult* questions. From Table 2, we observe a clear trend that skill-based examples harm an SLM's performance on the set of *easy* questions, while effectively boosting performance on the *difficult* ones. To gain deeper insight into how skill-based in-context examples might harm performance on *easy* questions, we present two illustrative cases where the model's performance regresses when using such prompts.

**Case Study 1: Skill-based examples lead the model to overlook key problem constraints.** In this example (see Appendix C.1), the Qwen2.5-7B-Instruct model is given an algebra question that includes multiple geometric constraints. When prompted with fixed examples, the model correctly identifies two possible answers and chooses the correct one according to the given condition "both coordinates are negative." On the other hand, when conditioned on examples that represent algebraic skills, the model overly emphasizes algebraic completeness but overlooks this important problem condition. It finally selects the incorrect answer by a random guess.

**Case Study 2: Symbol-heavy skill-based examples cause the model to overthink.** This question (see Appendix C.2) requires a plug-in-and-test approach instead of solving an equation. With fixed in-context examples, the model is able to find out the correct answer by directly plugging in and trying out small values. However, the skill-based examples that involve equation solving may have caused the model to overthink. After failing in the first plug-in-and-test, it ended up attempting to solve the equation system and eventually failed.

#### 4.1.1 Fine-grained Analysis: Effect of skill-based examples across five difficulty levels

The above observations motivate a more fine-grained analysis. We partition our evaluation set into five levels of difficulty, based on the probability of success under Best-of-$n$ sampling (Gui et al., 2024), verified using ground-truth labels. Formally, a question belongs to Difficulty Level $\ell$ ($1 \leq \ell \leq 4$) if it can be solved with Best-of-$2^{\ell-1}$ sampling, but not with any lower $n$. Questions that belong to Level 5 can't be solved with Best-of-8 sampling. We provide no in-context examples when measuring the success of Best-of-$n$ sampling and use temperature of 1.0. Intuitively, questions in Level 2 are those where the model is more susceptible to minor issues like formatting, where fixed in-context examples could help. For questions in higher levels, on the other hand, the model might benefit more from guidance with carefully selected in-context examples.

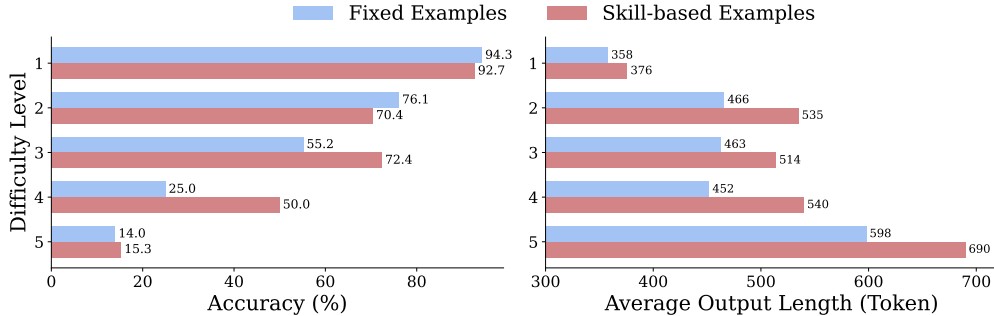

Figure 3: Accuracy and average output length of Qwen2.5-3B-Instruct on questions of Difficulty Level 1–5, designed using its Best-of-$n$ performance, with fixed and skill-based examples. Skill-based examples hinder performance on Levels 1 and 2, while helping on Levels 3–5. On all difficulty levels, skill-based examples result in noticeably longer outputs.

After splitting the questions into 5 levels, we compare the effect of skill-based in-context examples with fixed in-context examples on the model's responses to questions in each difficulty level. Figure 3 reports the results on a Qwen-3B model and MATH dataset.

**Primary observations:** We clearly observe that skill-based in-context examples can perform worse than fixed in-context examples in levels 1 and 2. On the other hand, skill-based in-context examples can substantially help the model on questions in levels 3–5. Furthermore, we observe that responses of the model are substantially longer with skill-based in-context examples, when compared with model responses with fixed in-context examples.

This shows that with skill-based examples, the model can return unnecessarily longer responses and make mistakes on easier questions, when simple strategies like Best-of-2 sampling or prompting with fixed in-context examples would have sufficed. This aligns with existing works on the issues of longer chain-of-thought reasoning in language models and how it relates to "problems of over-thinking" in humans (Liu et al., 2024b; Diaconis & Mazur, 2003). [2]

## 4.2 Ablation Studies

**Effect of in-context example choices in Stage 2.** Our main method combines *difficult* questions with skill-based examples and *easy* ones with fixed examples, based on the observation that models only need targeted instructions on more challenging cases. To better understand its effectiveness, we conduct an ablation study exploring alternative combinations of in-context examples. Our primary observations are

- As shown in Figure 4, our combination of "difficult+skill-based; easy+fixed" consistently outperforms all other configurations. Notably, the accuracy gap between the best and worst-performing combination can reach 7.1%, which stresses the importance of carefully choosing in-context examples for SLMs.

- The sensitivity to in-context example selection varies across model sizes, with the 1.5B model being the most sensitive and the 7B model being the most stable.

**Effect of threshold values on the reward model prediction.** We investigated the effect of $\tau_1$ and $\tau_2$ (defined in Section 2.2) on the classification performance of *easy* or *difficult* questions. Specifically, we measure whether our classification of questions as *easy* or *difficult* also corresponds to the correctness of responses assessed using ground-truth labels. In

---

[2]We also present results using the difficulty split of questions annotated in the original MATH dataset in Appendix B.3. Differences in performance and generation length of model's responses with skill-based and fixed in-context examples are less pronounced across difficulty levels. This is expected, as model's own responses must be a better fine-grained indicator on question difficulty.

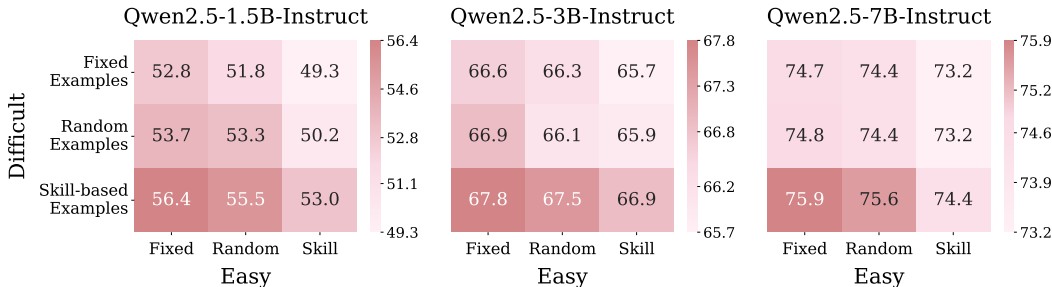

Figure 4: ICL performance, measured in terms of accuracy, across different combinations of in-context examples for *easy* and *difficult* questions on the MATH dataset. Across all models, we observe that skill-based in-context examples for *difficult* questions and fixed in-context examples for the *easy* questions work the best.

Table 3, we report four metrics (accuracy / precision / recall / F1) evaluating the prediction accuracy resulting from different filtering thresholds. Note that $\tau_1 = 0$ or $\tau_2 = 0$ means completely removing the constraints of $\tau_1$ or $\tau_2$. Across all evaluated combinations of threshold values, our choice of the threshold values ($\tau_1 = 0.85, \tau_2 = 0.7$) gives a good combination of prediction scores. To further visualize this effect, we conduct AdaptMI on top of all combinations of thresholds, and report the final accuracy in Table 4. Our choice of threshold values yields the highest final accuracy among all the combinations.

| $\tau_1 \backslash \tau_2$ | $\tau_2 = 0$ | $\tau_2 = 0.6$ | $\tau_2 = 0.7$ | $\tau_2 = 0.8$ |
|---|---|---|---|---|
| $\tau_1 = 0$ | 53 / 0 / 0 / 0 | 80 / 78 / 79 / 79 | 80 / 74 / 88 / 79 | 75 / 66 / 95 / 78 |
| $\tau_1 = 0.8$ | 80 / 79 / 78 / 79 | 80 / 76 / 85 / 80 | 79 / 72 / 90 / 80 | 75 / 66 / 96 / 78 |
| $\tau_1 = 0.85$ | 79 / 74 / 88 / 80 | 79 / 72 / 90 / 80 | **78 / 70 / 92 / 80** | 74 / 65 / 96 / 78 |
| $\tau_1 = 0.9$ | 73 / 64 / 95 / 77 | 73 / 64 / 95 / 77 | 72 / 64 / 96 / 77 | 70 / 62 / 97 / 75 |

Table 3: Reward model performance (accuracy / precision / recall / F1) on classifying correct/incorrect responses from Qwen2.5-1.5B-Instruct on MATH, across different thresholds. $\tau_1 = 0$ or $\tau_2 = 0$ means completely removing $\tau_1$ or $\tau_2$. Our choice of threshold values ($\tau_1 = 0.85, \tau_2 = 0.7$) gives a good combination of prediction scores.

| $\tau_1 \backslash \tau_2$ | $\tau_2 = 0$ | $\tau_2 = 0.6$ | $\tau_2 = 0.7$ | $\tau_2 = 0.8$ |
|---|---|---|---|---|
| $\tau_1 = 0$ | 52.8 | 55.7 | 55.9 | 55.7 |
| $\tau_1 = 0.8$ | 55.1 | 56.3 | 56.2 | 55.6 |
| $\tau_1 = 0.85$ | 55.3 | 56.4 | **56.4** | 55.6 |
| $\tau_1 = 0.9$ | 55.7 | 55.7 | 55.6 | 55.2 |

Table 4: Final AdaptMI performance of Qwen2.5-1.5B-Instruct on MATH, with different thresholds. Our choice of threshold values ($\tau_1 = 0.85, \tau_2 = 0.7$) leads to the highest accuracy.

**Additional ablations.** We compare a process reward model with an outcome reward model in Appendix B.1. We further show the potential of using alternate heuristic filtering methods to use in place of reward models to classify *easy* and *difficult* questions. We find that these heuristic strategies could replace reward models with appropriate hyperparameters. We keep full exploration to future work. We also explore an alternative strategy to construct adaptive in-context instruction, where we feed in natural language instructions provided by LLM in place of in-context examples, in Appendix B.2. We find that the models simply ignore in-context information that contain long, and unstructured natural language feedback.

## 5 Related Works

**In-context learning example selection.** As a key feature of language models, the in-context learning ability (Brown et al. (2020)) enables models to improve performance without undergoing gradient-based training. This ability can be maximally activated with carefully chosen in-context demonstrations. Prior works have extensively studied the dynamics of in-context learning (Chen et al. (2024)) and effective techniques of in-context example selection (Zhang et al. (2022); Cheng et al. (2023); An et al. (2023); Didolkar et al. (2024a); Liu et al. (2024a)) for larger models (>13B). These heuristics often simply rely on the semantic relation between the question and examples, and they typically require training a dedicated example selection model. Meanwhile, the in-context learning dynamics of small language models are understudied.

**Classifying model failures.** Identifying and understanding language model failures helps us adaptively improve model performance, e.g., via targeted training data selection (Zeng et al. (2025)). Prior works have utilized models' test-time failure patterns to build adaptive datasets with *difficult* questions (Dinan et al. (2019); Nie et al. (2020); Ribeiro & Lundberg (2022); Gao et al. (2023); Li et al. (2025)). However, these failure identification and classification approaches have rarely been applied to inform in-context example selection.

**Symbolic and Skill-based Reasoning.** Performing symbolic reasoning can largely enhance language models' math reasoning ability (Sullivan & Elsayed (2024); Alotaibi et al. (2024); Xu et al. (2024); Shaik & Doboli (2025)). As SLMs generally possess weaker capabilities to understand complex in-context information, symbolic knowledge aids SLM reasoning by providing structured, less-noisy contextual information (Liao et al. (2024)). Notably, the concept of "skill" was proven effective as a useful criterion for clustering symbolic knowledge (Didolkar et al. (2024a)), guiding contextual example selection (Didolkar et al. (2024a); An et al. (2023)) and mixture-of-experts routing (Chen et al. (2025)).

## 6 Conclusion

Our work explores reasons behind the failure of skill-based in-context examples to boost ICL performance of SLMs. We show that skill-based selection can make the model "overthink" on easier questions, which leads to a degradation in ICL performance. We then propose adaptive in-context selection strategies, AdaptMI and AdaptMI+, that use skill-based selection only for *difficult* questions.

While our primary focus is on improving ICL performance in SLMs, an important question is whether similar strategies can also guide the training of better SLMs. Current approaches often rely on distilling (Hinton et al., 2015) an SLM directly from the logits or generations of a frontier LLM, which requires careful curation of training data and training pipeline for optimal and efficient benefits (Hsieh et al., 2023; Ivison et al., 2023; Kaur et al., 2024). Recent studies suggest that additional in-context information can help models learn more effectively or efficiently. However, these strategies employ static or manually crafted curricula and in-context information (Zhu et al., 2025; Gao et al., 2025; Liao et al., 2024; Allen-Zhu & Li, 2024). An important open direction, thus, is how to adapt AdaptMI and AdaptMI+ to enable SLMs to train more effectively using frontier LLMs.

## Acknowledgements

We thank the members of Princeton Language and Intelligence for their helpful discussion and feedback. Sanjeev Arora and Abhishek Panigrahi are funded by NSF, Darpa, ONR, and Schmidt Foundation. Abhishek Panigrahi is a current Apple AIML scholar.

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

# Appendix

## Table of Contents

# A   Experimental Details

## A.1   Skill Annotation on MATH and GSM8K

As described in Section 3, we follow Didolkar et al. (2024a) to label skills on both the training and test sets of MATH and GSM8K using GPT-4o-mini (OpenAI, 2024). We enlist all skills that we used to annotate the questions in MATH and GSM8K dataset in Tables 6 and 7 and appendix A.1, which have been taken from Didolkar et al. (2024a). We ask the LLM to read the question and provide up to five skills required to solve this question, from the given existing skill list. We show an example prompt for annotating MATH Number Theory questions as follows.

---

**Example skill annotation prompt for MATH Number Theory questions**

[TASK]
You'll be given a math question. Your task is to output:
(1) < skill> list here up to five skill(s) that are required to solve this problem, seperated by commas </skill>.
(2) <reason> reason here why these skills are needed </reason>.

[SKILL LIST]
You should only choose the skills from this list:
[
"arithmetic_sequences",
"base_conversion",
"basic_arithmetic",
"division_and_remainders",
"exponentiation",
"factorization",
"greatest_common_divisor_calculations",
"modular_arithmetic",
"number_manipulation",
"number_theory",
"polynomial_operations",
"prime_number_theory",
"sequence_analysis",
"solving_equations",
"understanding_of_fractions"
]

[QUESTION]
{question}

[REASON AND SKILL(S)]

---

Table 5 shows some example MATH questions and their corresponding annotated skills. From the skill annotation, we construct a Skill Bank (see Figure 1 and Section 2.1) that stores the required skills for each question.

| Question | Annotated skills |
|---|---|
| What is the units digit of $3^1 + 3^3 + 3^5 + 3^7 + \ldots + 3^{2009}$? | exponentiation, modular arithmetic, sequence analysis |
| In the addition problem each letter represents a distinct digit. What is the numerical value of E? [Figure] | basic arithmetic, number manipulation, solving equations |
| In triangle $ABC$, $\tan(\angle CAB) = \frac{22}{7}$, and the altitude from $A$ divides $\overline{BC}$ into segments of length 3 and 17. What is the area of triangle $ABC$? | geometry and space calculation, trigonometric calculations, arithmetic operations |

Table 5: Example MATH questions, and the annotated skills generated by GPT-4o-mini.

| Subject | List of Skills |
|---|---|
| Per subject split in MATH | |
| Algebra | algebraic_expression_skills, algebraic_manipulation_skills, arithmetic_skills, calculation_and_conversion_skills, combinatorial_operations_and_basic_arithmetic, complex_number_skills, distance_and_midpoint_skills, exponent_and_root_skills, factoring_skills, function_composition_skills, function_skills, geometric_sequence_skills, graph_and_geometry_skills, inequality_skills, logarithmic_and_exponential_skills, number_theory_skills, polynomial_skills, quadratic_equation_skills, ratio_and_proportion_skills, sequence_and_series_skills, solving_equations |
| Counting and Probability | calculating_and_understanding_combinations, combinatorial_mathematics, combinatorics_knowledge, counting_principals, factorials_and_prime_factorization, number_theory_and_arithmetic_operations, permutation_and_combinations, probability_calculation_with_replacement, probability_concepts_and_calculations, probability_theory_and_distribution, understanding_and_applying_combinatorics_concepts |
| Geometry | 3d_geometry_and_volume_calculation_skills, algebraic_skills, area_calculation_skills, circle_geometry_skills, combinatorics_and_probability_skills, coordinate_geometry_and_transformation_skills, other_geometric_skills, pythagorean_skills, quadrilateral_and_polygon_skills, ratio_and_proportion_skills, triangle_geometry_skills, trigonometry_skills, understanding_circle_properties_and_algebraic_manipulation |

Table 6: List of skills used for annotating questions in each subject in MATH dataset

| Subject | List of Skills |
|---|---|
| | Per subject split in MATH |
| Intermediate Algebra | `absolute_value_skills, algebraic_manipulation_and_equations, calculus_optimization_skills, complex_number_manipulation_and_operations, function_composition_and_transformation, graph_understanding_and_interpretation, inequality_solving_and_understanding, polynomial_skills, properties_and_application_of_exponents, quadratic_equations_and_solutions, recursive_functions_and_sequences, sequence_and_series_analysis_skills, simplification_and_basic_operations, solving_inequalities, solving_system_of_equations, summation_and_analysis_of_series, understanding_and_application_of_functions, understanding_and_applying_floor_and_ceiling_functions, understanding_and_manipulation_of_rational_functions, understanding_and_utilizing_infininte_series, understanding_ellipse_properties, understanding_logarithmic_properties_and_solving_equations` |
| Number Theory | `arithmetic_sequences, base_conversion, basic_arithmetic, division_and_remainders, exponentiation, factorization, greatest_common_divisor_calculations, modular_arithmetic, number_manipulation, number_theory, polynomial_operations, prime_number_theory, sequence_analysis, solving_equations, understanding_of_fractions` |
| Pre-algebra | `average_calculations, basic_arithmetic_operations, circles, counting_and_number_theory, exponentiation_rules, fractions_and_decimals, geometry, multiples_and_zero_properties, multiplication_and_division, perimeter_and_area, prime_number_theory, probability_and_combinatorics, ratio_and_proportion, solving_linear_equation` |
| Pre-calculus | `algebra_and_equations, basic_trigonometry, calculus, complex_number_operations, complex_numbers, coordinate_systems, determinant_calculation, geometric_relations, geometry_and_space_calculation, geometry_triangle_properties, matrix_operations, parametric_equations, sequences_series_and_summation, three_dimensional_geometry, trigonometric_calculations, vector_operations` |

Table 7: List of skills used for annotating questions in each subject of MATH dataset (continued from Table 6)

---

**List of skills used for annotating questions in GSM8K**

`advanced_arithmetics, algebra, basic_arithmetics, conversion_unit_operations, division, equations, fractions, interest_calculations, multiplication, percentage, problem_solving, profit_loss, ratio_proportion, time_speed_distance`

---

### A.2 Missing skill Identification from Model Responses

As described in Section 2.3, we use GPT-4o-mini to label the skills that are missing from a model response. We ask the LLM to read the question along with the SLM response and provide the skills that the model fails to leverage in the response, from the given existing skill list. Below we show an example prompt for labeling missing skills for MATH Number Theory questions, as well as an example LLM output.

**Example missing skill identification prompt for a MATH Number Theory question**

[TASK]
You'll be given a math question and a step-by-step solution written by a Small Language Model. Your task is to output:
(1) $<$judge $>$ judge here whether the solution is correct or incorrect $</$judge $>$
(2) $<$reason $>$ if it's incorrect, reason here why the solution is incorrect $</$reason $>$,
(3) $<$skill $>$ list here what skill(s) should the SLM enhance in order to answer correctly, seperated by commas $</$skill $>$.

[SKILL LIST]
You should only choose the skills from this list:
[ "arithmetic_sequences",
"base_conversion",
"basic_arithmetic",
"division_and_remainders",
"exponentiation",
"factorization",
"greatest_common_divisor_calculations",
"modular_arithmetic",
"number_manipulation",
"number_theory",
"polynomial_operations",
"prime_number_theory",
"sequence_analysis",
"solving_equations",
"understanding_of_fractions"]

[QUESTION]
Given that $p \geq 7$ is a prime number, evaluate

$$1^{-1} \cdot 2^{-1} + 2^{-1} \cdot 3^{-1} + 3^{-1} \cdot 4^{-1} + \cdots + (p-2)^{-1} \cdot (p-1)^{-1} \pmod{p}.$$

[MODEL_SOLUTION]
To evaluate the sum $S = 1^{-1} \cdot 2^{-1} + 2^{-1} \cdot 3^{-1} + 3^{-1} \cdot 4^{-1} + \cdots + (p-2)^{-1} \cdot (p-1)^{-1}$ (mod $p$), where $p$ is a prime number greater than or equal to 7, we start by noting that for any integer $a$ not divisible by $p$, the modular inverse $a^{-1}$ is the integer $b$ such that $ab \equiv 1 \pmod{p}$. This means that $a \cdot a^{-1} \equiv 1 \pmod{p}$.

We can rewrite each term in the sum $S$ as follows: $a^{-1} \cdot (a+1)^{-1} \equiv (a+1) \cdot a^{-1} \cdot (a+1)^{-1} \equiv (a+1) \cdot a^{-1} \pmod{p}$.

Now, let's consider the sum $S$ more carefully. We can write: $S = \sum_{a=1}^{p-2} a^{-1} \cdot (a+1)^{-1} = \sum_{a=1}^{p-2} (a+1) \cdot a^{-1} \pmod{p}$.

This becomes $S = \sum_{a=1}^{p-2} \left(1 + \frac{1}{a}\right) = (p-2) + \sum_{a=1}^{p-2} a^{-1}$. It is known that $\sum_{a=1}^{p-1} a^{-1} \equiv 0 \pmod{p}$, so $\sum_{a=1}^{p-2} a^{-1} \equiv -(p-1)^{-1} \equiv -(p-1) \pmod{p}$, since $(p-1)^2 \equiv 1 \pmod{p}$. Therefore, $S \equiv (p-2) - (p-1) \equiv -1 \pmod{p}$.

Thus, the value of the sum is $\boxed{-1}$.

[REASON AND SKILL(S)]

> **Example GPT-4o-mini output for missing skill identification**
>
> (1) \<judge\> incorrect \</judge\>
> (2) \<reason\> The solution incorrectly simplifies the expression and misapplies properties of modular arithmetic. The solution claims that the sum of the inverses from 1 to ( p-1 ) is (-1 pmodp), which is incorrect. The correct approach involves recognizing that each term $(a^{-1} \cdot (a+1)^{-1})$ should be evaluated individually, and the sum should be calculated directly without assuming incorrect simplifications. The solution also fails to correctly apply the properties of modular inverses and does not properly evaluate the sum. \</reason\>
> (3) \<skill\> modular_arithmetic, number_theory, understanding_of_fractions \</skill\>

### A.3 Skill-based Example Retrieval

We outline our algorithm for retrieving in-context examples tailored to a specific set of skills. Leveraging the Skill-Map definition in Section 2.1, which annotates each question with its associated skills, we construct an inverse mapping called Example-Bank : Skill-Bank($\mathcal{Q}$) → $\mathcal{P}$. This map associates each skill $s$ with the subset of in-context examples in the pool $\mathcal{P}$ that are linked to $s$ according to Skill-Map. Given a question $q$ and a target skill set $K$, we retrieve in-context examples by randomly selecting one example from Example-Bank($s$) for each skill $s$ in $K$. The algorithm is given in Algorithm 1.

---
**Algorithm 1** Skill-based example retrieval

---
**Input:** List of skills $K = [k_1, ..., k_n]$ ($n \leq 5$)
**Output:** Selected 5-shot examples $E = [e_1, ..., e_5]$

1: $E \leftarrow []$
2: **if** $K$ is not empty **then**
3:       ▷ We allow an additional repeated in-context example for the first $5 - n$ skills
4:     **for** $i = 1$ to $5 - n$ **do**
5:         $E' \leftarrow$ Example-Bank($k_1$)
6:         **if** $E'$ is not empty **then**
7:             $e \leftarrow$ random_choice($E'$)
8:             $E \leftarrow E$ + [e]
9:         **end if**
10:     **end for**
11:
12:     **for** each $k$ in $K$ **do**
13:         $E' \leftarrow$ Example-Bank($k$)
14:         **if** $E'$ is not empty **then**
15:             $e \leftarrow$ random_choice($E'$)
16:             $E \leftarrow E$ + [e]
17:         **end if**
18:     **end for**
19: **end if**
20:
21:
22: $E \leftarrow Set(E)$         ▷ Remove repeated instances
23: **if** len($E$) < 5 **then**
24:     Append examples from fixed in-context examples to fill remaining shots
25:       ▷ This happens in the rarest of cases when we don't have enough examples for a skill!
26: **end if**
27: **return** $E$

---

# B Ablation Study

## B.1 Ablations on the reward filtering method in Stage 1

Recall that in Stage 1 of the AdaptMI pipeline, we use an off-the-shelf process reward model (RLHFlow/Llama3.1-8B-PRM-Mistral-Data) to score small language models' responses, in order to filter out a set of *difficult* questions for each model. Here, we conduct various ablation studies on the reward filtering process.

**Out-of-distribution (OOD) prediction performance of reward model.** Although we primarily evaluated AdaptMI on MATH and GSM8K, our method can potentially be extended to other math datasets. While the reward model we used in Stage 1 was only trained on the MATH and GSM8K distribution, we show that it is capable of scoring responses for various OOD math datasets. Table 8 reports the reward model's performance on classifying correct/incorrect responses from Qwen2.5-7B-Instruct on four popular math benchmarks: AMC23, AIME24, AIME25, and MATH[2]. The reward model achieves comparably high performance on scoring SLM responses on these OOD, significantly more difficult benchmarks, indicating that the model is highly generalizable. This implies the potential to extend our method to new datasets without the need to train a specialized reward model for each one.

| Metric | AMC23 | AIME24 | AIME25 | MATH[2] |
|---|---|---|---|---|
| Accuracy | 92.5 | 86.7 | 86.7 | 84.8 |
| Precision | 90.9 | 92.6 | 86.7 | 95.2 |
| Recall | 95.2 | 92.6 | 100.0 | 88.5 |
| F1 | 93.0 | 92.6 | 92.9 | 91.0 |

Table 8: Reward model prediction metrics across four OOD math benchmarks. Despite not being trained on these benchmarks, the reward model's prediction capability is largely generalizable to them.

**Reward Filtering vs. Simple Heuristics for classifying difficult questions.** Considering the computational overhead of calling a separate PRM, we explored alternative approaches to classifying questions that rely on computation-free simple heuristics. Specifically, we experimented with two heuristic strategies:

- **Consistency heuristic:** We measure the consistency of the model across five sampled generations per question and classify questions with lower consistency as difficult. Specifically, a question is *difficult* if, among 5 sampled generations, the most common response appears $< 2$ times.

- **Length heuristic:** We use the length of the model's responses as a proxy and classify questions with longer responses as difficult. Specifically, a question is *difficult* if the average model response length on this question is $\geq 800$ words.

Table 9 shows that both heuristics yield reasonably accurate predictions. Moreover, applying AdaptMI on top of these heuristic-classified difficult questions can improve the final accuracy by 2%. However, we leave a more thorough investigation into the robustness and generalizability of these strategies in relation to PRM-based classification for future work.

**Process Reward vs. Outcome Reward.** We also compare the prediction accuracy of our process reward model (PRM) with threshold filtering (see Section 2.2) against directly loading the reward model as an outcome reward model (ORM). Our preliminary experiments indicated 0.9 as the optimal threshold for the outcome rewards. With $\tau = 0.9$, the prediction metrics of the ORM are: Precision = 0.54 / Recall = 0.90 / F1 = 0.68, whereas the prediction metrics of the PRM with optimal thresholds are Precision = 0.70 / Recall = 0.92 / F1 = 0.80. Therefore, our method using PRM with threshold filtering is superior to directly using ORM.

| Classification method | Classification accuracy | SLM accuracy w/o AdaptMI → w/ AdaptMI |
|---|---|---|
| Consistency Heuristic | 79.80% | 52.8% → 54.8% **(+2.0%)** |
| Length Heuristic | 74.20% | 52.8% → 54.6% **(+1.8%)** |
| Reward Filtering | 78.00% | 52.8% → 56.4% **(+3.6%)** |

Table 9: Performance of consistency heuristic and length heuristic on classifying difficult questions. The classification accuracy of simple heuristics are on par with the reward filtering method. Applying Stage 2 of AdaptMI on top of the heuristic-classified difficult questions can yield improvement on the final accuracy by 2%.

## B.2 Comparing few-shot instructions with natural language instructions

Here, we explore an alternative strategy to construct adaptive in-context instruction. We want to test whether additional supervision from the LLM in AdaptMI+ could be provided in terms of *feedback* using natural language instructions.

For *difficult* questions, we modify our adaptive instruction as follows. After getting the predicted missing skills on model's response from an LLM, we prompt the LLM back with the missing skills and the corresponding skill-based in-context examples and ask the model to return a concise *natural language LLM feedback* that contains criticism on the model's response, and hints on how to apply the required skills. See below for an example prompt.

|  | MATH | GSM8K |
|---|---|---|
| Fixed Examples | 74.7 | 91.7 |
| LLM Feedback | 74.4 | 91.2 |
| + Fixed Examples | 74.5 | 91.1 |
| + Skill Examples | 74.5 | 91.1 |

Table 10: Qwen2.5-7B-Instruct accuracy under LLM-generated natural language instructions.

We report the behavior of modified AdaptMI+ on Qwen2.5-7B-Instruct. Interestingly, we observe that even 7B models tend to not benefit from the unstructured instructions (see Table 10). Furthermore, even if skill-based in-context examples are utilized along with LLM feedback, the SLM's performance remains nearly unchanged, which suggests the model simply ignores in-context information that contains long, and unstructured natural language feedback.

---

**Example feedback labeling prompt**

[TASK]
You'll be given a math question, a step-by-step solution written by a SLM, a step that is likely to be incorrect, the missing skills in the solution that you identified earlier, and the skill-rated in-context examples. Your task is to output:
(1) <comment> comment here about the SLM solution </comment>
(2) <hint> write here compactly the hints we should give the SLM to help it correctly answer this question next time. You should not include the question or answer of this specific question. </hint>

[QUESTION]
Square $ABCD$ has side length 2. A semicircle with diameter $\overline{AB}$ is constructed inside the square, and the tangent to the semicircle from $C$ intersects side $\overline{AD}$ at $E$. What is the length of $\overline{CE}$? [Figure]

[SOLUTION]
To solve the problem, we start by identifying the coordinates of the points in the square ABCD . Let's place the square in the coordinate plane with A = (0,0) , B = (2,0) , C = (2,2) , and D = (0,2) . The semicircle is constructed with diameter \overline{AB} , so its center is at the midpoint of \overline{AB} , which is (1,0) , and its radius is 1. The equation of the semicircle is (x-1)^2 + y^2 = 1 for y \geq 0 . We need to find the point E where the tangent from C to the semicircle intersects side \overline{AD} . Since E lies on \overline{AD} , its coordinates are (0, y_E) . The slope of the line from C to E is \frac{y_E - 2}{0 - 2} = \frac{2 - y_E}{2} . The equation of the line CE is: \[ y - 2 = \frac{2 - y_E}{2}(x - 2). \] Since E = (0, y_E) , substituting x = 0 and y = y_E into the equation, we get: \[ y_E - 2 = \frac{2 - y_E}{2}(-2) = y_E - 2. \] This confirms that the point E lies on the line. To find the exact coordinates of E , we use the fact that the tangent to the semicircle at a point (x_1, y_1) has the equation: \[ (x_1 - 1)(x - 1) + y_1 y = 1. \] Since the tangent passes through C = (2,2) , substituting x = 2 and y = 2 into the tangent equation, we get: \[ (x_1 - 1)(2 - 1) + y_1 \cdot 2 = 1 \implies x_1 - 1 + 2y_1 = 1 \implies x_1 + 2y_1 = 2. \] Also, since (x_1, y_1) lies on the semicircle, we have: \[ (x_1 - 1)^2 + y_1^2 = 1. \] We solve these two equations simultaneously. From x_1 + 2y_1 = 2 , we get x_1 = 2 - 2y_1 . Substituting this into the semicircle equation, we get: \[ (2 - 2y_1 - 1)^2 + y_1^2 = 1 \implies (1 - 2y_1)^2 + y_1^2 = 1 \implies 1 - 4y_1 + 4y_1^2 + y_1^2 = 1 \implies 5y_1^2 - 4y_1 = 0 \implies y_1(5y_1 - 4) = 0. \] So, y_1 = 0 or y_1 = \frac{4}{5} . Since y_1 = 0 corresponds to the point A , which is not a valid tangent point, we have y_1 = \frac{4}{5} . Then x_1 = 2 - 2 \cdot \frac{4}{5} = \frac{2}{5} . The point of tangency is \left( \frac{2}{5}, \frac{4}{5} \right) . The length of \overline{CE} is the distance from C = (2,2) to E = \left(0, \frac{4}{5}\right) : \[ CE = \sqrt{(2-0)^2 + \left(2 - \frac{4}{5}\right)^2} = \sqrt{4 + \left(\frac{10}{5} - \frac{4}{5}\right)^2} = \sqrt{4 + \left(\frac{6}{5}\right)^2} = \sqrt{4 + \frac{36}{25}} = \sqrt{\frac{100}{25} + \frac{36}{25}} = \sqrt{\frac{136}{25}} = \frac{2\sqrt{34}}{5}. \] Thus, the length of \overline{CE} is \boxed{\frac{2\sqrt{34}}{5}} .

[INCORRECT_STEP]
The slope of the line from C\) to E\) is \frac{y_E - 2}{0 - 2} = \frac{2 - y_E}{2}\). The equation of the line CE\) is: \[ y - 2 = \frac{2 - y_E}{2}(x - 2). \] Since E = (0, y_E)\), substituting x = 0\) and y = y_E\) into the equation, we get: \[ y_E - 2 = \frac{2 - y_E}{2}(-2) = y_E - 2. \] This confirms that the point E\) lies on the line.

[MISSING_SKILLS]
triangle_geometry_skills,circle_geometry_skills,algebraic_skills

[COMMENT_AND_HINT]

> **Example feedback from GPT-4o-mini**
>
> <comment>The SLM's solution is mostly correct but could benefit from a more detailed explanation of the triangle inequality and how it affects the possible values of the third side. Additionally, the SLM might have overlooked the strict inequality in the triangle inequality theorem, leading to the incorrect conclusion that the minimum value of BC is exactly 2x .</comment>
>
> <hint>
> - Emphasize the importance of the triangle inequality theorem, specifically a + b > c , where a , b , and c are the sides of a triangle.
> - Explain why BC must be strictly greater than 2x to satisfy the inequality BC + x > 3x .
> - Clarify that the smallest integer value for BC that satisfies this condition is 2x + 1 , leading to the calculation of the maximum possible height h .
> </hint>

### B.3 Fine-grained analysis of skill-based and fixed in-context examples on original manual split of MATH dataset

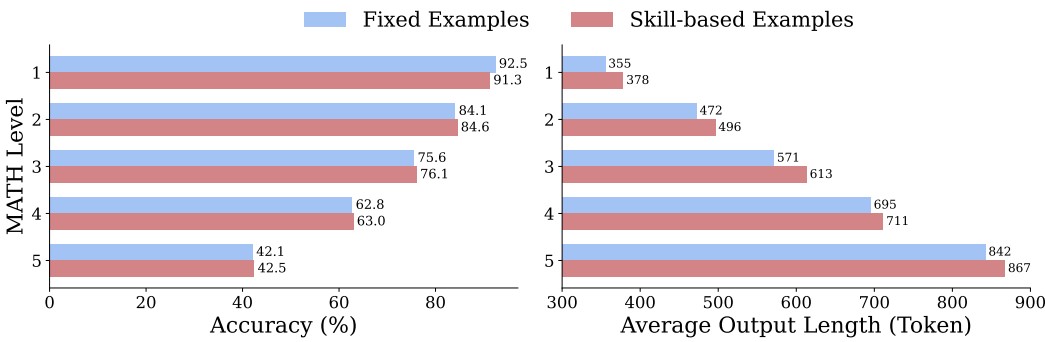

Figure 5: Accuracy and average output length of Qwen2.5-3B-Instruct on questions of Level 1–5 defined in the MATH dataset. Compared to Figure 3, the performance gap between fixed and skill-based examples is unnoticeable across all levels.

We repeat our experiment from Section 4.1.1. However, now instead of using Best-of-$n$ sampling to split the evaluation set into 5 levels, we use the manual split of questions given in the original MATH dataset. We report comparisons between skill-based and fixed in-context example selection strategies in Figure 5.

Interestingly, the differences between the ICL performance and generation length with skill-based and fixed in-context examples for the SLM are less pronounced across the 5 difficulty levels, compared to the results in Figure 3. This suggests that the manual difficulty split in the MATH dataset may not align well with the model's own perception of question difficulty. To capture more fine-grained distinctions between the two strategies, using the model's own responses through Best-of-$n$ sampling serves as a more reliable indicator of question difficulty.

## C Case Studies

In this section, we conduct case studies to gain deeper insight into how skill-based in-context examples might harm performance on easy questions, as mentioned in Section 4. We present two questions where SLM successfully solves with fixed examples, while failing with skill-based examples.

## C.1 Skill-based examples lead the model to overlook key problem constraints

In the example below, the Qwen2.5-7B-Instruct model is given an algebra question that includes multiple geometric constraints. While the question involves both Geometry and Algebra, it is only classified as an Algebra question in MATH, hence being combined with algebraic skill examples. When prompted with fixed examples, the model correctly identifies two possible answers and chooses the correct one according to the given condition "both coordinates are negative." On the other hand, when conditioned by examples that represent algebraic skills, the model overly emphasizes algebraic completeness but overlooks this important problem condition. It finally selects the incorrect answer by a random guess.

---

**Case Study 1: An example skill-based prompt**

< |im_start |>system
Please reason step by step, and put your final answer within \boxed{}.
< |im_end |>

< |im_start |>user
[Example 1 with skill "distance_and_midpoint_skills"]
[Example 2 with skill "distance_and_midpoint_skills"]
[Example 3 with skill "distance_and_midpoint_skills"]
[Example 4 with skill "algebraic_expression_skills"]
[Example 5 with skill "arithmetic_skills"]

A point $(x, y)$ on the coordinate plane with both coordinates negative is a distance of 6 units from the $x$-axis. It is a distance of 15 units from the point $(8, 3)$. It is a distance $\sqrt{n}$ from the origin. What is $n$?
< |im_end |>

< |im_start |>assistant

---

---

**Case Study 1: Correct model response under fixed example prompt**

To solve the problem, we need to determine the coordinates (x, y) of the point that satisfies the given conditions and then find the distance from the origin.

1. Distance from the x-axis:
The point (x, y) is 6 units away from the x-axis. Since the y-coordinate is negative, we have:
$$|y| = 6 \implies y = -6$$

2. Distance from the point $((8, 3))$:
The point $((x, y))$ is 15 units away from $((8, 3))$. Using the distance formula:
$$\sqrt{(x - 8)^2 + (y - 3)^2} = 15$$
....
This gives us two possible values for $(x)$:
$$x = 8 + 12 = 20 \quad \text{or} \quad x = 8 - 12 = -4$$

3. Distance from the origin:
We need to find the distance from the origin $((0, 0))$ to the point $((x, y))$. The distance formula is:
$$\sqrt{x^2 + y^2}$$
We have two possible points: $((20, -6))$ and $((-4, -6))$.

- For $((20, -6))$:
$$\sqrt{20^2 + (-6)^2} = \sqrt{400 + 36} = \sqrt{436}$$
- For $((-4, -6))$:
$$\sqrt{(-4)^2 + (-6)^2} = \sqrt{16 + 36} = \sqrt{52}$$

**Since the problem states both coordinates are negative, only $\sqrt{52}$ is valid**.

The correct value of $(n)$ is: n = 52

Thus, the final answer is: \boxed{52} ✅

---

**Case Study 1: Incorrect model response under skill-based example prompt**

To solve the problem, we need to determine the coordinates $((x, y))$ of the point that satisfies the given conditions and then find the distance from the origin.
1. Distance from the $(x)$-axis:

[Same as correct response]

2. Distance from the point $((8, 3))$:

[Long equation-solving process]

3. Distance from the origin:
We need to find the distance from the origin $((0, 0))$ to the point $((x, y))$. We have two possible points: $((20, -6))$ and $((-4, -6))$. - For $((20, -6))$: $$\sqrt{20^2 + (-6)^2} = \sqrt{400 + 36} = \sqrt{436} = 2\sqrt{109}$$ Thus, $(n = 436)$. - For $((-4, -6))$: $$\sqrt{(-4)^2 + (-6)^2} = \sqrt{16 + 36} = \sqrt{52} = 2\sqrt{13}$$ Thus, $(n = 52)$. **Since the problem does not specify which point to use**, we consider the first solution where $(n = 436)$.Therefore, the value of $(n)$ is $(\boxed{436})$. ✕

## C.2 Symbol-heavy skill-based examples cause the model to overthink.

The question below requires a plug-in-and-test approach instead of solving an equation. With fixed in-context examples, the model is able to find out the correct answer by directly plugging in and trying out small values. However, the skill-based examples that involve equation solving may have caused the model to overthink. After failing in the first plug-in-and-test, it ended up attempting to solve the equation system and eventually failed.

---

**Case Study 2: An example skill-based prompt**

< |im_start |>system
Please reason step by step, and put your final answer within \boxed{}.
< |im_end |>

< |im_start |>user
[Example 1 with skill "sequence_and_series_skills"]
[Example 2 with skill "sequence_and_series_skills"]
[Example 3 with skill "sequence_and_series_skills"]
[Example 4 with skill "function_composition_skills"]
[Example 5 with skill "solving_equations"]

Charlize accidentally omitted two consecutive integers when adding the elements of the arithmetic sequence, $\{1, 2, 3, \ldots, n\}$. If the sum she obtained is 241, what is the smallest possible value of $n$?
< |im_end |>

< |im_start |>assistant

---

**Case Study 2: Correct model response under fixed example prompt**

To solve the problem, we first need to determine the sum of the arithmetic sequence $\(\{1, 2, 3, \ldots, n\}\)$. The sum of the first $\(n\)$ positive integers is given by the formula: $\[ S = \frac{n(n+1)}{2} \]$

Let the two consecutive integers that Charlize omitted be $\(k\)$ and $\(k+1\)$. The sum of these two integers is: $\[ k + (k+1) = 2k + 1 \]$ Therefore, the sum of the sequence with these two integers omitted is: $\[ S - (2k + 1) = \frac{n(n+1)}{2} - (2k + 1) \]$ We know that this sum is 241, so we can write: $\[ \frac{n(n+1)}{2} - (2k + 1) = 241 \]$ Rearranging this equation, we get: $\[ \frac{n(n+1)}{2} = 242 + 2k \]$ Multiplying both sides by 2, we obtain: $\[ n(n+1) = 484 + 4k \]$ Since $\(k\)$ is an integer, $\(484 + 4k\)$ must be a product of two consecutive integers. We need to find the smallest $\(n\)$ such that $\(n(n+1)\)$ is close to 484 and is of the form $\(484 + 4k\)$.

**Let's check the values of $\(n\)$ starting from the smallest possible $\(n\)$ that is greater than $\(\sqrt{484} = 22\)$:**

1. If $\(n = 22\)$, then $\(n(n+1) = 22 \times 23 = 506\)$. We need to check if $\(506 = 484 + 4k\)$: $\[ 506 - 484 = 22 \implies 4k = 22 \implies k = \frac{22}{4} = 5.5 \]$ Since $\(k\)$ must be an integer, $\(n = 22\)$ is not a solution.
2. If $\(n = 23\)$, then $\(n(n+1) = 23 \times 24 = 552\)$. We need to check if $\(552 = 484 + 4k\)$: $\[ 552 - 484 = 68 \implies 4k = 68 \implies k = \frac{68}{4} = 17 \]$ Since $\(k = 17\)$ is an integer, $\(n = 23\)$ is a solution. The two consecutive integers omitted are 17 and 18. Therefore, the smallest possible value of $\(n\)$ is: $\[ \boxed{23} \]$

Thus, the final answer is: \boxed{23} ✅

---

---

**Case Study 2: Incorrect model response under skill-based example prompt**

To solve the problem, we first need to determine the sum of the arithmetic sequence $\{1, 2, 3, \ldots, n\}$. The sum of the first $n$ positive integers is given by the formula: $$S = \frac{n(n+1)}{2}$$

Charlize omitted two consecutive integers, say $k$ and $k+1$. Therefore, the sum she obtained is: $$S - (k + (k+1)) = S - (2k + 1) = \frac{n(n+1)}{2} - (2k + 1)$$ We know that this sum is 241, so we have: $$\frac{n(n+1)}{2} - (2k + 1) = 241$$ Rearranging the equation, we get: $$\frac{n(n+1)}{2} = 242 + 2k + 1 = 243 + 2k$$ Multiplying both sides by 2, we obtain: $$n(n+1) = 486 + 4k$$ Since $k$ is an integer, $486 + 4k$ must be a product of two consecutive integers. We need to find the smallest $n$ such that $n(n+1)$ is close to 486 and is of the form $486 + 4k$.

Let's check the values of $n$ starting from the smallest possible $n$ that makes $n(n+1) > 486$: For $n = 22$: $$n(n+1) = 22 \times 23 = 506$$ We need to check if $506$ can be written as $486 + 4k$: $$506 = 486 + 4k \implies 4k = 20 \implies k = 5$$ So, $n = 22$ works with $k = 5$. The two consecutive integers omitted are 5 and 6. Let's verify the sum: The sum of the sequence $\{1, 2, 3, \ldots, 22\}$ is: $$\frac{22 \times 23}{2} = 253$$ If we omit 5 and 6, the sum is: $$253 - (5 + 6) = 253 - 11 = 242$$ This is not correct.

**Instead, let's try to solve the equation:**

[Model attempts the solve the equation]

[Model reaches maximum output length before finishing] ✗

---

# D Additional Results

## D.1 Classification results of *easy* and *difficult* questions

In Stage 1 of AdaptMI (see Section 2.2), we identify a set of *difficult* questions for each individual model using a process reward model along with a filtering heuristic. Table 11 reports the proportions of difficult questions classified for different models in each math domain. Compared to Table 1, the proportions of difficult questions closely correspond to the accuracy numbers of each model, even though we did not access the ground truth in the whole pipeline. Notably, our classification method captures not only questions that the model gets wrong, but also questions that the model passes with a flawed solution process.

| Model | Geometry | Precalculus | Algebra | Prealgebra | Intermediate Algebra |
|---|---|---|---|---|---|
| Qwen2.5-1.5B-Instruct | 69.7 | 74.9 | 45.0 | 45.1 | 82.2 |
| Qwen2.5-3B-Instruct | 61.8 | 70.1 | 29.7 | 33.2 | 75.9 |
| Qwen2.5-7B-Instruct | 59.3 | 67.9 | 29.1 | 29.3 | 72.9 |
| Llama-3.2-1B-Instruct | 93.5 | 92.0 | 91.4 | 89.7 | 99.0 |
| Llama-3.2-3B-Instruct | 68.2 | 82.7 | 45.5 | 48.9 | 85.7 |

| Model | Count.&Prob. | Number Theory | MATH Avg. | GSM8K |
|---|---|---|---|---|
| Qwen2.5-1.5B-Instruct | 70.3 | 65.2 | 61.9 | 48.6 |
| Qwen2.5-3B-Instruct | 62.2 | 56.1 | 52.1 | 26.6 |
| Qwen2.5-7B-Instruct | 56.8 | 54.6 | 49.5 | 24.0 |
| Llama-3.2-1B-Instruct | 97.9 | 95.2 | 94.0 | 72.8 |
| Llama-3.2-3B-Instruct | 65.2 | 62.3 | 62.3 | 40.8 |

Table 11: Proportions of difficult questions (%) classified by AdaptMI for each model. Although our method did not access the ground truth, the proportion of classified difficult questions still closely mirrors each model's accuracy (see Table 1) in each domain.

### D.2 AdaptMI and AdaptMI+ performances

In addition to Table 1, we put the accuracy results on Number Theory, Intermediate Algebra, and Counting & Probability in MATH in Table 12. These results align with each other— AdaptMI and AdaptMI+ yield substantial improvement compared with all Pass@1 baseline, while being on par with the Consistency@5 results.

### D.3 Effect of skill-based examples on difficult and easy questions

In Section 4, we introduce our observation that skill-based examples only boost SLM performances on difficult questions but harm performance on easier ones. We present the additional results on Qwen2.5-3B-Instruct and Qwen2.5-7B-Instruct in Table 13 and Table 14. Similar to Table 2, there is a clear performance drop on easy questions with skill-based examples, although the drop for Qwen2.5-3B-Instruct and Qwen2.5-7B-Instruct is less significant than Qwen2.5-1.5B-Instruct.

| Methods | Number Theory | Intermediate Algebra | Counting & Probability |
|---|---|---|---|
| *# Qwen2.5-1.5B-Instruct* | | | |
| Fixed Examples | 45.2 | 36.5 | 47.3 |
| Random Examples | 43.7 | 35.1 | 47.3 |
| Skill-based Examples | 45.4 | 35.8 | 44.7 |
| Consistency@5 | **50.0** | **39.8** | 47.8 |
| **AdaptMI** | 49.8 | 36.9 | 50.0 |
| **AdaptMI+** | 49.1 | 38.4 | **51.5** |
| *# Qwen2.5-3B-Instruct* | | | |
| Fixed Examples | 65.9 | 46.8 | 59.5 |
| Random Examples | 64.1 | 46.7 | 60.1 |
| Skill-based Examples | 66.1 | 45.9 | 60.3 |
| Consistency@5 | 66.5 | 49.4 | 61.7 |
| **AdaptMI** | 66.7 | 46.5 | 60.6 |
| **AdaptMI+** | **68.9** | **49.8** | **62.7** |
| *# Qwen2.5-7B-Instruct* | | | |
| Fixed Examples | 74.8 | 57.3 | 72.6 |
| Random Examples | 74.4 | 55.7 | 73.4 |
| Skill-based Examples | 73.0 | 55.9 | 71.1 |
| Consistency@5 | **79.1** | 57.5 | 71.7 |
| **AdaptMI** | 73.5 | 57.6 | 71.5 |
| **AdaptMI+** | 77.4 | **58.8** | **74.9** |
| *# Llama-3.2-1B-Instruct* | | | |
| Fixed Examples | 10.3 | 7.8 | 11.5 |
| Random Examples | 7.3 | 7.9 | 6.9 |
| Skill-based Examples | 11.2 | 7.3 | 10.4 |
| Consistency@5 | **21.4** | 6.7 | **14.3** |
| **AdaptMI** | 12.1 | 7.8 | 12.5 |
| **AdaptMI+** | 10.3 | **8.9** | 13.5 |
| *# Llama-3.2-3B-Instruct* | | | |
| Fixed Examples | 38.7 | 22.6 | 42.7 |
| Random Examples | 29.3 | 18.5 | 33.7 |
| Skill-based Examples | 39.6 | 23.2 | 33.7 |
| Consistency@5 | 35.0 | 21.1 | **46.7** |
| **AdaptMI** | **43.4** | **24.4** | 39.3 |
| **AdaptMI+** | **43.4** | **24.4** | 39.3 |

Table 12: Additional results of Table 1. AdaptMI and AdaptMI+ also demonstrate consistent accuracy gain compared with baseline methods. All results are Pass@1 accuracy unless otherwise indicated. **Exp.** stands for Examples. The selection methods for fixed, random, and skill-based examples are introduced in Section 2.1

| Question & Example | | MATH | | | | |
|---|---|---|---|---|---|---|
| | | Geometry | Precalculus | Algebra | Prealgebra | Number Theory |
| # Qwen2.5-1.5B-Instruct | | | | | | |
| Diff. | Fixed | 21.3 | 23.7 | 44.8 | 35.1 | 24.1 |
| | Random | 23.2 | 25.3 | 53.9 | 40.5 | 21.9 |
| | Skill | 28.4 | 28.9 | 55.1 | 45.5 | 31.2 |
| | | +7.1 | +5.2 | +10.3 | +10.4 | +7.1 |
| Easy | Fixed | 82.1 | 81.8 | 94.6 | 93.7 | 84.6 |
| | Random | 81.6 | 78.9 | 92.1 | 92.3 | 80.1 |
| | Skill | 77.2 | 71.5 | 85.9 | 86.0 | 71.8 |
| | | -4.9 | -10.3 | -8.7 | -7.7 | -12.8 |
| # Qwen2.5-3B-Instruct | | | | | | |
| Diff. | Fixed | 36.5 | 37.9 | 60.6 | 48.1 | 49.5 |
| | Random | 36.8 | 38.7 | 62.6 | 50.5 | 49.2 |
| | Skill | 34.1 | 41.8 | 68.3 | 54.3 | 50.8 |
| | | -2.4 | +3.9 | +7.7 | +6.2 | +1.3 |
| Easy | Fixed | 88.5 | 90.2 | 95.9 | 95.4 | 86.9 |
| | Random | 83.6 | 86.5 | 94.0 | 94.0 | 84.1 |
| | Skill | 84.7 | 88.3 | 93.8 | 93.8 | 85.7 |
| | | -3.8 | -1.8 | -2.2 | -1.6 | -1.3 |
| # Qwen2.5-7B-Instruct | | | | | | |
| Diff. | Fixed | 50.0 | 51.3 | 80.1 | 71.6 | 66.8 |
| | Random | 48.3 | 52.5 | 81.3 | 71.3 | 67.3 |
| | Skill | 52.0 | 57.4 | 81.5 | 74.7 | 66.9 |
| | | +2 | +6.1 | +1.4 | +3.1 | +0.1 |
| Easy | Fixed | 90.8 | 93.9 | 98.7 | 97.5 | 93.3 |
| | Random | 92.6 | 93.4 | 99.2 | 97.7 | 91.9 |
| | Skill | 89.8 | 91.4 | 96.0 | 94.7 | 91.5 |
| | | -1.0 | -2.5 | -2.7 | -2.8 | -1.8 |

Table 13: Accuracy of Qwen2.5-1.5B-Instruct, Qwen2.5-3B-Instruct, and Qwen2.5-7B-Instruct on *difficult* and *easy* questions, respectively under fixed, random, and skill-based examples (additional results for Table 2). Skill-based examples boost performance on *difficult* questions across all categories, while significantly underperforming on *easy* questions. The gap between easy and difficult questions is more pronounced for smaller models.

| Question & Example | | Intermediate Algebra | MATH
Counting & Probability | Avg. | GSM8K
Avg. |
|---|---|---|---|---|---|
| # Qwen2.5-1.5B-Instruct | | | | | |
| Diff. | Fixed | 27.0 | 28.2 | 29.8 | 45.2 |
| | Random | 23.0 | 27.2 | 31.2 | 46.1 |
| | Skill | 27.4 | 32.1 | 35.7 | 48.0 |
| | | +0.4 | +3.9 | +5.9 | +2.8 |
| Easy | Fixed | 80.7 | 92.2 | 90.2 | 96.3 |
| | Random | 75.7 | 88.1 | 87.6 | 90.6 |
| | Skill | 74.5 | 74.5 | 81.0 | 83.2 |
| | | -6.2 | -17.7 | -9.2 | -13.1 |
| # Qwen2.5-3B-Instruct | | | | | |
| Diff. | Fixed | 35.5 | 40.3 | 42.9 | 51.6 |
| | Random | 36.2 | 41.0 | 43.4 | 56.7 |
| | Skill | 35.0 | 42.0 | 45.2 | 61.8 |
| | | -0.4 | +1.69 | +2.3 | +10.2 |
| Easy | Fixed | 82.6 | 91.1 | 92.4 | 96.7 |
| | Random | 81.8 | 90.9 | 91.8 | 95.6 |
| | Skill | 79.8 | 90.5 | 90.4 | 93.9 |
| | | -2.8 | -0.6 | -2.0 | -2.8 |
| # Qwen2.5-7B-Instruct | | | | | |
| Diff. | Fixed | 50.0 | 61.5 | 60.7 | 74.1 |
| | Random | 49.1 | 62.7 | 60.8 | 76.7 |
| | Skill | 51.2 | 61.9 | 62.7 | 77.0 |
| | | +1.2 | +0.4 | +2 | +2.9 |
| Easy | Fixed | 89.7 | 96.1 | 96.2 | 97.3 |
| | Random | 86.5 | 97.1 | 95.3 | 96.4 |
| | Skill | 86.1 | 94.7 | 94.1 | 95.5 |
| | | -3.6 | -1.4 | -2.1 | -1.8 |

Table 14: Accuracy of Qwen2.5-1.5B-Instruct, Qwen2.5-3B-Instruct, and Qwen2.5-7B-Instruct on *difficult* and *easy* questions, respectively under fixed, random, and skill-based examples (additional results for Table 2). Skill-based examples boost performance on *difficult* questions across all categories, while significantly underperforming on *easy* questions. The gap between easy and difficult questions is more pronounced for smaller models.

