# OpenReview forum: "AdaptMI: Adaptive Skill-based In-context Math Instructions for Small Language Models"
_colmweb.org/COLM/2025/Conference — COLM 2025_

### Official Review · Reviewer_nCJe · 2025-05-11

**Rating:** 7
**Confidence:** 3
**Ethics Flag:** 1

**Summary:**

This paper aims to improve the in-context learning (ICL) performance of Small Language Models (SLMs). It proposes AdaptMI, a method for selecting skill-based in-context Math Instructions for SLMs. An enhanced approach, AdaptMI+ is further proposed to identify the key skills missing in the SLM’s responses for difficult questions. The evaluation is conducted using the MATH and GSM8K datasets. The results show that both AdaptMI and AdaptMI+ methods outperform the baseline methods.

**Ethics Concerns Details:**

Null

**Questions To Authors:**

Concerning the evaluation presented in Table 2 and Figure 3, could the authors elaborate on the rationale for not including a comparison between methods AdaptMI and AdaptMI+ and the LLM?

For the results shown in Table 1, the AdaptMI+ method did not consistently outperform AdaptMI. An explanation or analysis of the factors contributing to these instances would be appreciated.

**Reasons To Accept:**

The structure of the paper is clear. The study presents a detailed experimental design and a thorough analysis of the findings.

**Reasons To Reject:**

Null.

---

> ### Author Response · Authors · 2025-06-02
>
> We thank the reviewer for their positive comments on our work. Please find responses to your questions below.
>
> ### **“An explanation or analysis of the factors contributing to the instances, where AdaptMI+ method did not consistently outperform AdaptMI”**
>
> **Response:** We thank the reviewer for the question. As the only difference between AdaptMI+ and AdaptMI is the skill identification strategy (AdaptMI+ used an LLM to identify missing skills, while AdaptMI directly retrieved with skills from the Skill Bank), it is hard to statistically quantify when a different set of selected skill-based examples contributes to a better/worse performance.
>
> To give a small example, we conducted a small-scale case study on the questions where the small model initially failed, got corrected with AdaptMI, but again failed with AdaptMI+. A common pattern observed in such cases is that AdaptMI+ tends to overfit to the small model’s previous mistake, rather than addressing the broader skill required to solve the question.
>
> For instance, in this question (in MATH “Precalculus”):
>
> > *Find the spherical coordinates of the point diametrically opposite $\\left( 3, \\frac{3 \\pi}{8}, \\frac{\\pi}{5} \\right).$*
>
> In this case, AdaptMI correctly identified the relevant skill as three_dimensional_geometry, highlighting the need to reason over all three spherical coordinates. In contrast, AdaptMI+ identified the missing skill as geometry_and_space_calculation, which narrowly targeted the model's previous mistake and failed to generalize to the full scope of the question. As a result, the small model failed again. We believe a deeper analysis on the differences between AdaptMI and AdaptMI+ can be an interesting future direction.
>
> ### **“Could the authors elaborate on the rationale for not including a comparison between methods AdaptMI and AdaptMI+ and the LLM?”**
>
> **Response:** Below we present the comparison between our approaches and the LLM performance (gpt-4o-mini). On GSM8K, the LLM is on par with Qwen2.5-7B-Instruct. While on MATH, the LLM is only on par with the 3B model, lacking behind the 7B model by ~7%.
>
>
> |  | **Qwen2.5-1.5B-Instruct** |      **Qwen2.5-1.5B-Instruct**                | **Qwen2.5-3B-Instruct** |           **Qwen2.5-3B-Instruct**           | **Qwen2.5-7B-Instruct** |         **Qwen2.5-7B-Instruct**              |  |
> |--------------|---------------------------|----------------------|--------------------------|----------------------|--------------------------|----------------------|------------------|
> |      **Accuracy**         | **AdaptMI**               | **AdaptMI+**         | **AdaptMI**              | **AdaptMI+**         | **AdaptMI**              | **AdaptMI+**         |                  **gpt-4o-mini (LLM)** |
> | **MATH**     | 56.4                      | 57.2                 | 67.8                     | 69.1                 | 75.9                     | **76.7**             | 69.1             |
> | **GSM8K**    | 72.9                      | 75.8                 | 87.4                     | 87.7                 | 92.3                     | 92.4                 | **94.1**         |
>
> We did not include this comparison in the earlier paper version because our approach only relied on minimal, high-level outputs (i.e., providing skill labels) from the LLM, instead of distilling its reasoning capabilities. The performance gain of AdaptMI mainly stems from its adaptive nature.

---

### Official Review · Reviewer_Qc6H · 2025-05-13

**Rating:** 6
**Confidence:** 4
**Ethics Flag:** 1

**Summary:**

In this paper, the authors propose AdaptMI and an improved version AdaptMI+ for adaptive skill-based example selection in in-context learning. The methods contain two-stage: first detect the difficult and easy questions based on rewarding the inference LLM’s response; second select examples that correspond to the question skills (for AdaptMI) or missed skills (for AdaptMI+). Experimental results on several Small Language Models (SLM) demonstrate that these methods outperform existing baselines. Moreover, the authors provide detailed discussions to validate that SLM is likely to “overthinking” on easy questions given skills.

**Questions To Authors:**

1. In the construction of the Skill-Map, each question is associated with exactly $k$ skills. How do the authors ensure that every question indeed requires exactly $k$ distinct skills?
2. Since the proposed methods heavily rely on the predictions of a large LLM (GPT-4o-mini) for both skill annotation and missing skill identification, it would be important to understand how reliable these predictions are. Could the authors provide a quantitative analysis of the skill prediction accuracy, ideally benchmarked against human-annotated ground truth?
3. In Section 3.2, the authors state: "This may suggest that higher-performing models require a more intelligent and target skill identification process." I am a little confused, because the same skill identification LLM (GPT-4o-mini) is applied to all model backbones regardless of their performance level. How can the experimental results support the claim that higher-performing models need more refined skill selection?
4. Given that the core idea of AdaptMI/AdaptMI+ is to treat easy and difficult questions differently, it would be helpful to explicitly report the proportions of easy versus difficult questions as determined by the methods on each dataset.

**Reasons To Accept:**

1. This paper is well written, with a clear and logical presentation of both the proposed methods and experimental setup.
2. The authors provide a thorough and insightful analysis of how skill-based in-context examples affect the performance of SLMs. Their observations (e.g., such examples can introduce unnecessary cognitive load on easy problems) are well supported by empirical results.

**Reasons To Reject:**

1. The paper contains some overclaims. For instance, in the Introduction, the authors claim that AdaptMI+ "creates examples" for the missing skills. However, in implementation, they simply select examples from an existing annotated pool based on those missing skills—similar to how skill-based examples are selected for all predicted skills. The term “create” misleadingly implies the generation or synthesis of new content for the missing skills, which is not supported by the implementation details.
2. Certain technical details lack clarity (please refer to Q1 below), and there are notational inconsistencies, such as the variable $k$ representing both the number of required skills per question in Section 2.1 and the number of steps in a solution in Section 2.2.
3. The overall efficiency of the proposed methods is limited. They rely on training an additional process reward model and using a large LLM to predict the SLM’s missing skills, which introduces high computational overhead and potential error accumulation.
4. The logic behind AdaptMI+ is somewhat questionable. When using a large LLM to predict the missing skills, it implicitly assumes or requires that the large LLM can accurately solve the problem itself. However, if the large model can already solve the problem correctly, then AdaptMI+ is essentially providing the SLM with information that reflects the correct solution. It is akin to having a stronger model solve the problem first and then offering its reasoning path as a reference to the SLM. This makes me feel that the improved performance of AdaptMI+ is to be expected, but not due to a technical advancement, rather because it is indirectly leveraging the reasoning results of the larger model.
5. Some important experimental results require further explanation and visualization (please refer to Q2-4 below).

---

> ### Author Response · Authors · 2025-06-03
>
> We thank the reviewer for their careful review and feedback. Please find our responses to the concerns raised in the review.
>
> ### 1. **“High computational overhead in AdaptMI+ due to training an additional process reward model and using a large LLM”**
>
> We answer the question in two parts:
>
> **Re: Overhead due to training an additional process reward model (PRM)**
>
> In our experiments, the PRM was used to classify easy and difficult questions for the small model based on its responses. As mentioned in line 108, this was done to avoid access to ground truth labels. The PRM that we use is a general purpose model that is applicable to a wide range of math questions beyond MATH and GSM8K, even though it was trained on MATH and GSM8K questions. This indicates the potential to extend our method to new datasets without the need to train a specialized PRM for each one.
>
> |Metric|AMC23|AIME24|AIME25|MATH${ }^2$|
> |----------|-----|------|------|------|
> |Accuracy|92.5|86.7|86.7|84.8|
> |Precision|90.9|92.6|86.7|95.2|
> |Recall|95.2|92.6|100.0|88.5|
> |F1|93.0|92.6|92.9|91.0|
>
> **Table A:** PRM performance for Qwen2.5-7B-Instruct
>
> **Motivated by the reviewer’s question,** we also explored alternative approaches to classify questions as easy or difficult without relying on the PRM model. Specifically, we experimented with two heuristic strategies: (a) using the length of the model’s responses as a proxy and classifying questions with longer responses as difficult, and (b) measuring the consistency of the model across five sampled generations per question and classifying questions with lower consistency as difficult. Preliminary results suggest that both methods yield reasonably accurate predictions. However, we leave a more thorough investigation into the robustness and generalizability of these strategies in relation to PRM-based classification for future work.
>
> |Heuristic|Easy Definition|Prediction Accuracy|Accuracy of SLM (Fixed Examples)|Accuracy of SLM  with AdaptMI on heuristic easy/difficult split|
> |-------------|-------------------|------------------------|-----------------------------|-----------------------------------|
> |Consistency|Most common response appears ≥ 2 times|79.80%|52.8%|54.8%|
> |Length|Response length < 800 words|74.20%|52.8%|54.6%|
>
> **Table:** Performance of Qwen2.5-1.5B-Instruct on MATH
>
>
> **Re: Overhead of using an LLM**
>
> First, in AdaptMI, we use an LLM to label the skills required for each question and construct a corresponding skill bank. This labeling is performed once offline, introducing only a limited overhead. Despite this minimal reliance on the LLM, we demonstrate that it can lead to substantial improvements in the performance of the SLM.
>
> Building on this, we proposed AdaptMI+, an enhanced strategy where the LLM is additionally used to annotate the missing skills in the model’s responses. While we acknowledge that this approach incurs greater overhead due to repeated LLM usage, it also highlights that such iterative access can provide meaningful benefits for model performance.
>
>  ### 2. **“Improvement is expected, …, but not due to a technical advancement, rather because it is indirectly leveraging the reasoning results of the larger model”**
>
> Our framework is inherently a teacher-student setup, where we want a stronger model to improve the performance of a weaker model. This setup will only become more prominent over time, as we build stronger AI models in the future and leverage them to improve the weaker models.
>
> Secondly, we show that the teacher doesn’t need to give the ground truth answers directly to the student. Instead, we only utilize the metacognitive ability of the LLM to predict very high-level information about questions.
> - Below is an example of the extracted LLM output:
> > algebraic_skills, understanding_circle_properties_and_algebraic_manipulation, coordinate_geometry_and_transformation_skills
>
> The LLM is solely giving out skills without solving the problem or giving additional information.
> - There are only 118 skills for MATH and 14 for GSM8K, indicating that these skills are high-level instructions rather than a strong guidance for problem solving.
>
> **More importantly,** the teacher model (GPT-4o) is not necessarily stronger: it achieves only 69% on MATH, 5% lower than the baseline performance of Qwen2.5-7B-Instruct, and comparable to Qwen2.5-3B-Instruct. We will include this discussion in the next version.

---

> > ### Author Response · Authors · 2025-06-03
> >
> > ### 3. **“How can the experimental results support the claim that higher-performing models need more refined skill selection?”**
> >
> > Our claim is based on the behavior of different models with AdaptMI and AdaptMI+. Please note that AdaptMI+ uses an LLM to identify missing skills based on the SLM’s responses. Thus, we consider it to be a more intelligent and targeted approach as the skills are identified based on the model's test-time performance. For stronger models like Qwen 3B/7B, AdaptMI yields ~1% gain, whereas AdaptMI+ improves accuracy by ≥3%. In contrast, smaller models like LLaMA 1.5B already see strong gains with AdaptMI alone. This supports our claim that in order to improve the performance of high performing models, we need more refined skill selection method like AdaptMI+.
> >
> >
> > ### 4. **“Proportions of easy versus difficult questions as determined by the methods on each dataset.”**
> >
> > We thank the reviewer for the suggestion. Below in Table B, we attach the table with proportions of difficult questions for all the dataset domains. Compared to the model accuracy in Table 1, the proportion of difficult questions roughly mirrors the proportion of questions the model actually gets wrong.
> >
> >
> > |Model|Geometry|Precalculus|Algebra|Prealgebra|Intermediate Algebra|Counting and Probability|Number Theory|MATH Avg.|GSM8K|
> > |-----|--------|------------|-------|-----------|---------------------|-------------------------|--------------|----------|------|
> > |Qwen2.5-1.5B-Instruct|69.7|74.9|45.0|45.1|82.2|70.3|65.2|61.9|48.6|
> > |Qwen2.5-3B-Instruct|61.8|70.1|29.7|33.2|75.9|62.2|56.1|52.1|26.6|
> > |Qwen2.5-7B-Instruct|59.3|67.9|29.1|29.3|72.9|56.8|54.6|49.5|24.0|
> > |Llama-3.2-1B-Instruct|93.5|92.0|91.4|89.7|99.0|97.9|95.2|94.0|72.8|
> > |Llama-3.2-3B-Instruct|68.2|82.7|45.5|48.9|85.7|65.2|62.3|62.3|40.8|
> >
> > **Table B:** Proportions of difficult questions (%) on MATH and GSM8K for each model.
> >
> > ### 5. **“Quantitative analysis of the skill prediction accuracy, ideally benchmarked against human-annotated ground truth.”**
> >
> > First, we clarify that the annotated skills used in our work are generated by a large language model (LLM), not humans. Conducting a human study is particularly challenging due to the abstract and high-level nature of these skills, which function primarily as feedback signals exchanged between language models. Verifying whether a model's error can be attributed to a specific LLM-provided skill is a labor-intensive and nontrivial task.
> >
> > Instead, we present a preliminary investigation into the missing skill predictions in AdaptMI+ using an LLM-as-a-judge approach. We first evaluate GPT-4o-mini’s ability to self-verify the correctness of its own predicted missing skills and find that it judges its predictions to be correct 70% of the time. To further assess the reliability of these predictions, we compute the agreement between GPT-4o-mini and Claude-3.5-Sonnet. The models agree on 43% of the predicted skills, where agreement is defined as the average fraction of overlapping skills relative to the total number of skills predicted by GPT-4o-mini.

---

> > ### Author Response · Authors · 2025-06-03
> > **Clarifying questions on notations and wordings**
> >
> > ### **“In the construction of the Skill-Map, each question is associated with exactly k skills. How do the authors ensure that every question indeed requires exactly k distinct skills?”**
> >
> > We would like to clarify that the definition of Skill-Map used a fixed k notation to minimize notation overhead and simplify the discussion. In our experiments, one question is mapped to either 2 or 3 skills, as given by the LLM. We will modify it in the next version.
> >
> > ### **“The term “create” misleadingly implies the generation or synthesis of new content for the missing skills”**
> >
> > By ‘create’ in line 63, we meant ‘select’ in-context examples. We will modify wording in the next version.
> >
> > ### **“variable k representing both the number of required skills per question in Section 2.1 and the number of steps in a solution in Section 2.2.”**
> >
> > Thank you for pointing this out, we will use different variables in the next version.

---

> > > ### Comment · Reviewer_Qc6H · 2025-06-06
> > > **Response to authors**
> > >
> > > I appreciate the authors for providing very detailed responses to my questions. My questions and concerns have been addressed. Essentially, many of our discussions should be included in the revision to show the importance of this work. I would like to increase my score, and I hope the authors could pay more attention to adding the necessary clarifications within the rebuttal phase.

---

> > > > ### Author Response · Authors · 2025-06-06
> > > >
> > > > We sincerely thank the reviewer for their thoughtful feedback and kind words. We are glad that our responses have addressed the concerns raised. We will ensure that the key points discussed during the review are clearly incorporated into the revised version!

---

### Official Review · Reviewer_qoNx · 2025-05-15

**Rating:** 7
**Confidence:** 4
**Ethics Flag:** 1

**Summary:**

This paper uses an adaptive strategy to improve the efficacy of in-context learning (ICL) in SLMs, which are widely-known not to generalize well when given few-shot examples, specifically in the context of maths problems. This adaptive strategy relies on “skill-based prompting”, involving an intentional selection of ICL examples based on the difficulty of the question and the skill required to solve the question, to aid the model in improving its output.

**Questions To Authors:**

- In the partition of easy and difficult questions, you use $\tau_2$ as a lower bound of the reward value of any step. However, there could exist a case where the reward in one step could be low, but the average reward of the question is high (implying it is easy).  What would happen if we ablated $\tau_2$ entirely? How does it affect the model’s overall performance? Would like to see more variation in the threshold values to observe this

**Reasons To Accept:**

- Clear organization of the different sections; the paper flows well and is understandable to the reader

- Good methodological considerations on what works best to optimize performance and why (through ablation studies)

- Identified important gap in the literature on SLM and extrapolated their findings in the specific context of maths problems to a larger scope, pushing for adaptive learning strategies

- AdaptMI and use of skill-based problems in ICL is well-justified in the discussion, and further supported by the fine-grained analysis by difficulty

**Reasons To Reject:**

- Language use is anthropomorphic in some instances (e.g., “more likely to overthink”), which comes off as inaccurate, and the reference to cognitive theories as motivation for the learning strategy is bit surface-level and could be detailed further

- Section 3.2: only analyzes instances where AdaptMI and AdaptMI+ do better, could expand further on specific cases where consistency@5 does better/is on par with their approach and why to use one over the other (e.g., efficiency)

- Although skill-based prompting is properly justified for difficult questions, it is not clear that fixed examples are necessarily optimal for easy questions as performance is usually on par with randomly-selected examples.

---

> ### Author Response · Authors · 2025-06-02
>
> We thank the reviewer for their extensive feedback on our work. Please find our responses to your questions below.
>
> ### 1. **“Additional ablations on $\tau_2$ and its effect on the model’s overall performance”**
>
> **Response:** We thank the reviewer for the question. Below are ablation studies on the effect of varying $\tau_2$ or $\tau_1$ on small language models’ performance.
>
> As presented in Table A, entirely removing $\tau_2$ (i.e., set $\tau_2=0$) will indeed increase the prediction accuracy and precision, yet it decreases the recall and F1, which are indicators of the detection rate of model failures. Therefore, in Table B below, the model performances under $\tau_2=0$ turn worse. We observe the same trend on removing $\tau_1$.
> We will include these observations in our paper revisions.
>
> | $\tau_1 \backslash \tau_2$ | $\tau_2 = 0$     | $\tau_2 = 0.6$   | $\tau_2 = 0.7$   | $\tau_2 = 0.8$   |
> |-----------------------------|--------------------|--------------------|--------------------|--------------------|
> | $\tau_1 = 0$              | 53 / 0 / 0 / 0      | 80 / 80 / 78 / 79  | 80 / 74 / 88 / 79  | 75 / 66 / 95 / 78  |
> | $\tau_1 = 0.8$           | 80 / 79 / 78 / 79   | 80 / 76 / 85 / 80  | 79 / 72 / 90 / 80  | 75 / 66 / 96 / 78  |
> | $\tau_1 = 0.85$           | 79 / 74 / 88 / 80   | 79 / 72 / 90 / 80  | 78 / 70 / 92 / 80  | 74 / 65 / 96 / 78  |
> | $\tau_1 = 0.9$            | 73 / 64 / 95 / 77   | 73 / 64 / 95 / 77  | 72 / 64 / 96 / 77  | 70 / 62 / 97 / 75  |
>
> **Table A:** Reward model prediction metrics (accuracy / precision / recall / F1) across different thresholds for Qwen2.5-1.5B-Instruct on MATH.
>
>
> |$\tau_1 \backslash \tau_2$ | $\tau_2 = 0$ | $\tau_2 = 0.6$ | $\tau_2 = 0.7$ | $\tau_2 = 0.8$ |
> |-----------------------------|----------------|------------------|------------------|------------------|
> | $\tau_1 = 0$              | 52.8           | 55.7             | 55.9             | 55.7             |
> | $\tau_1 = 0.8$            | 55.1           | 56.3             | 56.2             | 55.6             |
> | $\tau_1 = 0.85$           | 55.3           | 56.4             | **56.4**         | 55.6             |
> | $\tau_1 = 0.9$            | 55.7           | 55.7             | 55.6             | 55.2             |
>
> **Table B:** Final performance of Qwen2.5-1.5B-Instruct on MATH under different thresholds for AdaptMI.
>
>
> ### 2. **“Specific cases where consistency@5 does better/is on par with their approach and why to use one over the other”**
>
> **Response:** Thank you for the suggestion. We will add the following content in Section 3.2:
>
> “While AdaptMI+ surpasses consistency@5 performance on most domains, it slightly lags behind on certain subjects such as Geometry and Precalculus for 1B or 3B models. These subjects are relatively difficult for the model, as suggested by their loss scores compared to other subjects (please check Table 9 in the appendix). Since AdaptMI requires models to have sufficient capabilities to leverage the given skill-based examples, it may not work better than Consistency@5 on these harder topics.”
>
> ### 3. **“It is not clear that fixed examples are necessarily optimal for easy questions as performance is usually on par with randomly-selected examples”**
>
>
> On easy questions, we acknowledge that fixed examples usually only result in 1~2% advantages over random examples. However, on some domains (e.g., Intermediate Algebra and Counting & Probability, see Table 10 in Appendix D.2), randomly-selected examples can lead to 5% performance drop compared to fixed examples on the Qwen 1.5B model.
> In terms of average performance on the whole dataset, Figure 4 shows that using fixed examples on easy questions consistently outperformed random examples. Therefore, we believe that fixed examples are the optimal choice for easy questions in the scope of our study.
>
>
> ### 4. **“the reference to cognitive theories as motivation for the learning strategy is bit surface-level and could be detailed further”**
>
> Thank you for the suggestion. The cognitive theories primarily serve as a motivational lens for studying adaptive in-context feedback in SLMs. We will include additional details and experiments in the next revision around our references to overthinking and adaptive teaching. We also refer the reviewer to our response 4 to Reviewer HGMM for related experiments on overthinking.

---

> > ### Comment · Reviewer_qoNx · 2025-06-05
> >
> > Thanks for resolving my comments, I don't have any further questions!

---

### Official Review · Reviewer_HGMM · 2025-05-16

**Rating:** 8
**Confidence:** 4
**Ethics Flag:** 1

**Summary:**

The paper mainly explores strategies of selecting examples for in-context learning (ICL), to improve the performance of smaller language models (SLMs, around 1B-7B parameters) in solving mathematical problems. Since different questions require different sets of *skills*, one possible strategy is to select examples that demonstrate related skills. Counterintuitively, this skill-based selection leads to worse performance for SLMs. The ablation study shows that the skill-based examples primarily benefit more difficult questions, but cause confusion for easier ones, possibly due to overthinking. The paper also proposes an adaptive method (AdaptiveMI) to address this issue. First, an external reward model judges the performance of the SLM, and opts in skill-based ICL examples only for challenging questions. Alternatively, a reflective LLM enhances the selection by picking skills the SLM fails to demonstrate (AdaptiveMI+). These methods improve performance to Consistency@5 level.

**Questions To Authors:**

## The overthinking hypothesis

The examples demonstrated to the SLMs are probably beyond their reasoning capability, therefore the ICL encourages overthinking. What about adding some statistics about these examples and the (over)thinking they triggered? In addition to Fig 3, how long are these examples, compared to outputs of different difficulty level? Further, what do you think of an ablation using ICL examples rewritten to each SLM’s own “reasoning style”?

**Reasons To Accept:**

I agree with the authors’ point on L264 that “the in-context learning dynamics of SLMs are understudied”. By intuition, the skill-based selection can be interpreted as using the most relevant “task” for ICL, which should outperform showing less relevant or random “tasks”. But the results in this paper clearly indicates that such examples can negatively impact the performance on questions that are already easy. This is an interesting insight to the broader community.

Besides, I think the paper is clearly written and easy to follow. One specific point I like is that instead of reusing the difficulty label within the MATH dataset, the experiments in Section 4 use difficulty labels derived from the SLM’s real performance. Difficulties judged by humans do not necessarily correspond with LM performance, yet the community seems to be less aware of this, so I feel this choice of difficulty metrics is also valuable.

**Reasons To Reject:**

## Method Effectiveness

The AdaptiveMI method uses much larger models (for AdaptiveMI+, GPT-4o-mini is used, which is probably an LLM already) to enhance SLMs under skill-based ICL example selection. Surely this method improves the performance to Consistency@5 level, but isn’t it requiring more computation than Consistency@5, because of all the extra larger models? Therefore, it might be less economic as Consistency@5 in real scenarios, especially for AdaptiveMI+. Another bottleneck of this method is the SkillBank, whose construction seems to be impossible with SLMs only.

## Additional Experiments

Putting effectiveness aside, I think there could have been more experiments to better explain the negative effect of skill-based ICL. Since the difficulty of a question is based on LM’s performance, statistically analyzing the correspondence of the difficulty and the performance impact of the skill-based ICL should be possible. Also it might be a good addition to show the distribution of difficulty level within each MATH subtask.

---

> ### Author Response · Authors · 2025-06-03
>
> We would like to thank the reviewer for the careful review and their positive feedback on our study. Please find our responses to your questions below.
>
> ### **“Computational bottlenecks, due to requiring additional models like PRM and LLM.”**
>
> We refer the reviewer to responses 1  to reviewer Qc6H. In summary, we address overhead in two parts:
>
> (1) the process reward model (PRM) is a general-purpose classifier trained once and applicable across math datasets, with alternatives like response length heuristics showing promising accuracy (up to 79.8%).
>
> (2) AdaptMI relies on a one-time LLM labeling step to create the skill bank and is effective in itself.  With additional overhead through repeated LLM use, AdaptMI+ can improve further.
>
> ### **“SkillBank’s construction seems to be impossible with SLMs only”**
>
> Our method uses a teacher-student framework where a relatively stronger model provides high-level skill annotations (not answers) based on metacognitive abilities. The teacher need not outperform the student; it only needs to be sufficiently capable of offering abstract skill-level guidance. In fact, in our response 2 to reviewer Qc6H, we show that our LLM scores 69% on MATH, below Qwen2.5-7B-Instruct.  Thus, while we did not explore building SkillBanks with SLMs to keep our scope focused, we believe it an important direction to consider for future work.
>
> ### **“Ablation on using ICL examples rewritten to each SLM’s own reasoning style”**
>
> We conducted this ablation study on Qwen models of different scales. We:
>
> (1) Prompted Qwen2.5-7B-Instruct to rewrite all the solutions in MATH by itself. We find that only the 7B model is capable of preserving key information while rewriting the solutions.
>
> (2) Replaced all the in-context examples with the original questions and rewritten solutions.
>
> (3) Tested the baselines and AdaptMI performance on all three models with the re-written ICL solutions.
>
> As shown in table below, rewriting examples yields a ~1% accuracy gain for the 1.5B model, suggesting it struggles more with human-written examples. However, the gains are minimal for larger models.
>
> |Accuracy on MATH|Qwen2.5-7B-Instruct||Qwen2.5-3B-Instruct||Qwen2.5-1.5B-Instruct||
> |----------------|-------------------|------------------|-------------------|------------------|---------------------|------------------|
> ||Original|Rewritten|Original|Rewritten|Original|Rewritten|
> |Fixed examples|74.7|**74.8**|66.6|**66.8**|52.8|**53.6**|
> |Skill-based examples|**74.4**|74.3|**66.9**|66.1|53.0|**54.1**|
> |AdaptMI|**75.9**|75.8|**67.8**|67.6|**56.4**|**57.1**|
>
>
> ### **“Statistics about length of skill-based examples and relation to (over)thinking they triggered”**
>
> First, we’d like to clarify that the fixed examples and skill-based examples are sampled from the same data distribution, i.e., the MATH or GSM8K training set. Hence, their average difficulty (and average length) should be similar to the output responses.
>
> For easy questions in Level 1 and 2, the skill-based examples have similar lengths as fixed examples on average. Therefore, the models’ overthinking on easy questions is less likely to come from long skill-based examples. Rather, it may be caused by skill-based examples being too specific to one topic, as exemplified in our case studies in Section 4.1 (line 200-213).
>
> More interestingly, we observe that for more difficult questions, their corresponding skill-based examples tend to be longer. This observation is likely because difficult questions often correspond to more advanced skills, such as solving equations or complex number calculation, and the associated ICL examples of these skills are longer.
>
> |Level|Number of questions|Fixed examples|Skill-based examples|With fixed examples|With skill-based examples|
> |-----|-------------------|---------------|---------------------|--------------------|--------------------------|
> |Level 1|249|107|103|358|376|
> |Level 2|47|107|106|466|535|
> |Level 3|37|107|116|463|514|
> |Level 4|59|107|135|452|540|
> |Level 5|108|107|140|598|690|
>
>
> ### **“Distribution of difficulty level (as judged by the PRM) within each MATH subtask”**
>
> Below, we show a table with proportions of difficult questions for all the dataset domains. The proportion of difficult questions roughly mirrors (though slightly exceeding) the proportion of questions the model actually gets wrong (table 1 in main paper) in each domain.
>
> |Model|Geometry|Precalculus|Algebra|Prealgebra|Intermediate Algebra|Counting and Probability|Number Theory|MATH Avg.|GSM8K|
> |-----|--------|------------|-------|-----------|---------------------|-------------------------|--------------|----------|------|
> |Qwen2.5-1.5B-Instruct|69.7|74.9|45.0|45.1|82.2|70.3|65.2|61.9|48.6|
> |Qwen2.5-3B-Instruct|61.8|70.1|29.7|33.2|75.9|62.2|56.1|52.1|26.6|
> |Qwen2.5-7B-Instruct|59.3|67.9|29.1|29.3|72.9|56.8|54.6|49.5|24.0|
> |Llama-3.2-1B-Instruct|93.5|92.0|91.4|89.7|99.0|97.9|95.2|94.0|72.8|
> |Llama-3.2-3B-Instruct|68.2|82.7|45.5|48.9|85.7|65.2|62.3|62.3|40.8|

---

> > ### Comment · Reviewer_HGMM · 2025-06-04
> >
> > Thanks for your detailed comment and extra ablations! I think these extra results have resolved my questions and I don't have more for now.

---

### Official Review · Reviewer_8zxM · 2025-05-20

**Rating:** 6
**Confidence:** 5
**Ethics Flag:** 1

**Summary:**

The paper introduces two methods AdaptMI and AdaptMI+ for selection of in-context skill based examples to promote in context learning in small language models for math datasets. They specifically observe that vanilla skill based examples hurt small language models on easy questions due to unnecessary information and hence require carefully chosen examples. Their methods show significant improvements in accuracy in some areas like pre-algebra.

**Questions To Authors:**

+ Why does skill based examples hurt SLM performance on easy questions? Can the "unnecessary information" or "cognitive overload" quantified ?
+ Is there anything specific to pre-training of SLM that affects the performance of in-context skill based examples.
+ Looks like most gains are made in pre-algebra and the model struggles to make significant gains in geometry ? Why is that ?

**Reasons To Accept:**

+ The paper provides a simple yet powerful method for selection of relevant examples for in-context learning.
+ They make clear observations on the performance of small language models on fixed set of examples and skill based examples

**Reasons To Reject:**

+ The authors do not provide details of the reward model used
+ The authors have also not provided a skill wise breakdown of performance of SLM: for example:  Which skill-based examples resulted in most improvement in accuracy?

---

> ### Author Response · Authors · 2025-06-02
>
> We thank the reviewer for their detailed feedback on our work. Please find responses to your questions below.
>
> ### **“No details of the reward model”**
>
> **Response:**  We would like to clarify that, in Section 3.1 (line 151-153), we specified the name, size, and training data of the reward model used, along with the prediction thresholds. Due to the space constraint, we put ablation studies on the reward model in Appendix B.1, where we different threshold values and compare process reward and outcome reward models. Please also check our responses to reviewer qoNx, where we add additional ablations on the effect of the thresholding functions. We will include them in our revision.
>
> ### **“Which skill-based examples resulted in the most improvement in accuracy?”**
>
> **Response:**  As recommended by the reviewer, we conducted a detailed analysis to identify the skill categories where skill-based in-context examples are most beneficial. We grouped the questions based on their annotated skills and measured the average performance improvement of a small language model (SLM) when switching from fixed to skill-based in-context examples, separately for easy and difficult questions. For Qwen2.5-1.5B-Instruct model, we find that questions involving the skill *"Perimeter and Area"* exhibit the greatest performance gain for difficult instances. In contrast, for questions requiring the skill *"Circle"*, performance notably declines for both easy and difficult variants.
>
> |**"Prealgebra" Skill Category** | **Easy questions** | **Difficult questions** |
> |----------------------------|------------------|------------------------|
> |Perimeter and Area|-22%|**+100%**|
> |Solving Linear Equation|+5%|+86%|
> |Average Calculations|-15%|+75%|
> |Multiples and Zero Properties|-6%|+75%|
> |Probability and Combinatorics|+0%|+60%|
> |Ratio and Proportion|-6%|+46%|
> |Multiplication and Division|-8%|+31%|
> |Basic Arithmetic Operations|-2%|+26%|
> |Geometry|-18%|+25%|
> |Fractions and Decimals|-13%|+18%|
> |Exponentiation Rules|-5%|+10%|
> |Counting and Number Theory|-12%|+9%|
> |Prime Number Theory|-24%|+0%|
> |Circles|**-40%**|**-67%**|
>
>
>
>
> ### **“Can the "unnecessary information" or "cognitive overload" be quantified ?”**
>
> **Response:** In Figure 3, we showed that small models are cognitive overloaded by skill-based examples on easy questions, as their output lengths were much longer, but accuracy dropped.
> To understand “why” skill-based examples brought cognitive overload, we propose two explanations:
>
> 1) The skill-based examples are too long, which confuses the model. (This hypothesis was also raised by reviewer HGMM.)
>
> 2) The skill-based examples are too specific to one topic, which misleads the model or causes overthinking.
>
> We already presented some case studies that indicate (2), in Section 4.1 (line 200-213).
> To test hypothesis (1), we calculated the average lengths of fixed examples and skill-based examples for questions in each difficulty level in Table B below.
>
> We observe that for more difficult questions, their corresponding skill-based examples tend to be longer. Yet for easy questions in Level 1 and 2, the skill-based examples have similar lengths as fixed examples on average. Therefore, the models’ overthinking on easy questions is less likely to come from long skill-based examples. Rather, the cognitive overload may be caused by skill-based examples being too specific to one topic. We believe that this effect is harder to quantify and is best explained by our case studies in Section 4.1.
>
> |**Level** |**Number of questions** | **Example length (token)** | **Example length (token)** | **Output length(token)** | **Output length (token)**|
> |-------|-------------------|-----------------------|-----------------------|----------------------|----------------------|
> |       |                   |**Fixed examples**|**Skill-based examples**|**With fixed examples**|**With skill-based examples**|
> |Level1|249|107|103|358|376|
> |Level2|47|107|106|466|535|
> |Level3|37|107|116|463|514|
> |Level4|59|107|135|452|540|
> |Level5|108|107|140|598|690|
>
> **Table B:** Average lengths of fixed examples and skill-based examples, in comparison with average output lengths of Qwen2.5-3B-Instruct, for questions in each difficulty level.
>
> ###  **“Is there anything specific to pre-training of SLM that affects the performance of in-context skill based examples”**
>
> **Response:** The varied performance of different SLMs with AdaptMI and AdaptMI+ suggests that pre-training influences how models use skill-based in-context examples. However, without access to detailed pre-training information on the models, predicting the exact factors is difficult.
>
> ###  **“Why are most gains in pre-algebra and the model struggles to make significant gains in geometry?”**
>
> **Response:** This likely reflects differences in pre-training; models may have seen more pre-algebra content than geometry. However, without detailed knowledge of the pre-training data, the exact cause is difficult to determine.

---

### Decision · Program_Chairs · 2025-07-08

**Decision:**

Accept

**Comment:**

Pros:
- This paper presents a technically strong method and its variation, AdaptMI and AdaptMI+, that significantly improves in-context learning accuracy in small language models.
- Systematic analysis has been offered.

All reviewers agree this paper is above the bar of being presented at COLM.

Cons that have been addressed in rebuttal and should be included in revision:
- Additional analysis (break-down analysis, ablation on $\tau_2$), as listed in the rebuttal
- If space allows, please also consider adding some future work discussions to include content that the authors are not exactly sure during the rebuttal, e.g.,
> “Why are most gains in pre-algebra and the model struggles to make significant gains in geometry?”
Response: This likely reflects differences in pre-training; models may have seen more pre-algebra content than geometry. However, without detailed knowledge of the pre-training data, the exact cause is difficult to determine.

Cons that should be addressed in revision:
- Please reduce the use of anthropomorphic languages. I am in agreement with Reviewer qoNx, while I acknowledge that might have somehow become a community convention, it is always encouraged to avoid potential misleading to the public. And that's our responsibility as scientists.

[Automatically added comment] At least one review was discounted during the decision process due to quality]